# Host-induced bacterial cell wall decomposition mediates pattern-triggered immunity in Arabidopsis

Xiaokun Liu[1†], Heini M Grabherr[1†], Roland Willmann[1], Dagmar Kolb[1], Frédéric Brunner[1], Ute Bertsche[2], Daniel Kühner[2], Mirita Franz-Wachtel[3], Bushra Amin[4], Georg Felix[1], Marc Ongena[5], Thorsten Nürnberger[1*], Andrea A Gust[1*]

[1]Department of Plant Biochemistry, Center for Plant Molecular Biology, University of Tübingen, Tübingen, Germany; [2]Department of Microbial Genetics, University of Tübingen, Tübingen, Germany; [3]Proteome Center Tübingen, University of Tübingen, Tübingen, Germany; [4]Medical and Natural Sciences Research Centre, University of Tübingen, Tübingen, Germany; [5]Wallon Centre for Industrial Biology, University of Liege-Gembloux Agro-Bio Tech, Gembloux, Belgium

**Abstract** Peptidoglycans (PGNs) are immunogenic bacterial surface patterns that trigger immune activation in metazoans and plants. It is generally unknown how complex bacterial structures such as PGNs are perceived by plant pattern recognition receptors (PRRs) and whether host hydrolytic activities facilitate decomposition of bacterial matrices and generation of soluble PRR ligands. Here we show that *Arabidopsis thaliana*, upon bacterial infection or exposure to microbial patterns, produces a metazoan lysozyme-like hydrolase (lysozyme 1, LYS1). LYS1 activity releases soluble PGN fragments from insoluble bacterial cell walls and cleavage products are able to trigger responses typically associated with plant immunity. Importantly, *LYS1* mutant genotypes exhibit super-susceptibility to bacterial infections similar to that observed on PGN receptor mutants. We propose that plants employ hydrolytic activities for the decomposition of complex bacterial structures, and that soluble pattern generation might aid PRR-mediated immune activation in cell layers adjacent to infection sites.

*For correspondence: nuernberger@zmbp.uni-tuebingen.de (TN); andrea.gust@zmbp.uni-tuebingen.de (AAG)

†These authors contributed equally to this work

## Introduction

Activation of antibacterial defenses in multicellular eukaryotic organisms requires recognition of bacterial surface patterns through host-encoded pattern recognition receptors (PRRs) (*Chisholm et al., 2006*; *Jones and Dangl, 2006*; *Ishii et al., 2008*; *Boller and Felix, 2009*; *Vance et al., 2009*; *Segonzac and Zipfel, 2011*; *Monaghan and Zipfel, 2012*; *Broz and Monack, 2013*; *Stuart et al., 2013*). Immunogenic microbial signatures are collectively referred to as pathogen- or microbe-associated molecular patterns (PAMPs/MAMPs) (*Janeway and Medzhitov, 2002*). Bacteria-derived PAMPs such as lipopolysaccharides (LPS) or flagellins possess immunity-stimulating activities in metazoans and plants, suggesting that the ability to sense bacterial surface structures and mount immunity is conserved across lineage borders (*Nürnberger et al., 2004*; *Boller and Felix, 2009*).

Likewise, peptidoglycans (PGNs) are major building blocks of the cell walls of Gram-positive and Gram-negative bacteria that have been shown to trigger host immune responses in mammalians, insects, and plants (*Dziarski and Gupta, 2005*; *Gust et al., 2007*; *Erbs et al., 2008*; *Kurata, 2014*). Structurally, PGNs are heteroglycan chains that are composed of polymeric alternating β(1,4)-linked N-acetylglucosamine (GlcNAc) and N-acetylmuramic acid (MurNAc) residues (*Schleifer and Kandler,*

**eLife digest** The immune response of plants and animals is triggered when cells detect small molecules that are present on the surface of the bacteria or fragments of peptidoglycans—the polymers that are a major component of the bacterial cell wall. The mechanisms by which small molecules trigger the immune response in plants have been widely studied in the model plant *Arabidopsis thaliana*, but less is known about the ways in which peptidoglycan fragments can initiate an immune response.

Proteins called lysozymes are known to break peptidoglycans into smaller pieces in animals. Plants do not produce lysozymes, but they do produce other enzymes such as chitinases that have similar properties. Now Liu, Grabherr, et al. have shown that a chitinase called LYS1 acts as an enzyme that catalyzes the breakdown of peptidoglycans and has a central role in triggering the immune response of Arabidopsis.

Plants that were genetically engineered to produce little or no LYS1 were highly susceptible to bacterial infection because there were no enzymes that could break the peptidoglycans into smaller fragments. However, plants that were engineered to produce very high levels of LYS1 also had a compromised immune response because the peptidoglycans were broken into fragments that were too small to be detected.

The findings of Liu, Grabherr et al. demonstrate that animals and plants employ similar strategies to break down bacterial peptidoglycans to allow them to be detected by the immune system. However, as the enzymes responsible have different structures, they are likely to have evolved separately in plants and animals.

*1972*; *Glauner et al., 1988*). Such chains are interconnected by oligopeptide bridges which form a coordinate meshwork, thereby providing structural integrity to the bacterial envelope. Recognition of different PGN substructures in animal hosts is brought about by structurally diverse PRRs such as nucleotide-binding oligomerization domain-containing proteins (NODs), peptidoglycan recognition proteins (PGRPs/PGLYRPs), scavenger receptors, or Toll-like receptor TLR2 (*Strober et al., 2006*; *Royet and Dziarski, 2007*; *Dziarski and Gupta, 2010*; *Müller-Anstett et al., 2010*; *Magalhaes et al., 2011*; *Kurata, 2014*). In plants, a tripartite PGN recognition system at the plasma membrane of *Arabidopsis thaliana* with shared functions in PGN sensing and transmembrane signaling was recently described (*Willmann et al., 2011*). This system comprises Lysin motif (LysM) domain proteins LYM1 and LYM3 for PGN ligand binding and the transmembrane LysM receptor kinase CERK1 that is likely required for conveying the extracellular signal across the plasma membrane and for initiating intracellular signal transduction. All three proteins were shown to be indispensable for PGN sensitivity and to contribute to immunity to bacterial infection (*Willmann et al., 2011*), which is in agreement with their proposed role as a PGN sensor system. More recently, a similar PGN perception system made of LysM domain proteins LYP4 and LYP6 has been reported from rice (*Liu et al., 2012a*).

Microbial patterns such as bacterial PGN, LPS, flagellin, or fungal chitin harbor immunogenic epitopes that are parts of supramolecular structures building microbial surfaces (*Boller and Felix, 2009*; *Kumar et al., 2013*; *Newman et al., 2013*; *Pel and Pieterse, 2013*). It is therefore assumed that recognition by host PRRs most likely requires the presence of soluble, randomly structured ligands derived from a complex matrix. X-ray structure-based insight into the binding of bacterial flagellin to the Arabidopsis receptor complex FLS2/BAK1 or of fungal chitin to the Arabidopsis receptor CERK1 supports this view (*Willmann and Nürnberger, 2012*; *Liu et al., 2012b*; *Sun et al., 2013*). Moreover, the existence of fungal LysM effector proteins that scavenge soluble chitin fragments, thus preventing recognition by plant PRRs, suggests that mechanisms releasing these soluble fragments from fungal cell walls must exist (*de Jonge et al., 2010*). Most often, however, it is an open question whether soluble ligand presentation to eukaryotic host PRRs is the result of spontaneous decomposition of the microbial extracellular matrix during infection or, alternatively, whether host-derived factors contribute to the generation of immunogenic ligands for PRR activation. For example, only monomers of bacterial flagellin induce immune responses through human TLR5 whereas filamentous flagella, in which the immunogenic flagellin structure is buried and thus is not accessible to TLR5, do not (*Smith et al., 2003*). It was proposed that a number of circumstances cause flagellin monomer release from intact flagella. For

instance, *Caulobacter crescentus* deliberately ejects its flagellum once it is no longer required for the bacterial life cycle (*Jenal and Stephens, 2002*). Moreover, during infection, *Pseudomonas aeruginosa* produces rhamnolipids which act as surfactants and cause flagellin shedding from intact flagella, resulting in a more pronounced immune response (*Gerstel et al., 2009*). Alternatively, host factors such as proteases or environmental conditions such as pH, temperature, or bile salts have been proposed to mediate shearing of flagella from bacterial surfaces (*Ramos et al., 2004*). Likewise, recognition of PGN by intracellular receptors, such as mammalian NOD1 and NOD2, or by plasma membrane receptors, such as mammalian TLR2 or plant LYM1, LYM3 and CERK1 (*Müller-Anstett et al., 2010*; *Sorbara and Philpott, 2011*; *Willmann et al., 2011*), is facilitated by soluble ligands. Animal lysozymes have been implicated in PGN hydrolysis, bacterial lysis, and host immunity (*Callewaert and Michiels, 2010*), probably through partial PGN degradation and generation of soluble ligands for PGN sensors (*Cho et al., 2005*; *Dziarski and Gupta, 2010*; *Davis et al., 2011*).

In plants, knowledge of the mode of release of immunogenic fragments from microbial extracellular structures and their contribution to plant immunity is lacking. We here describe a plant enzyme activity (LYS1) that hydrolyzes β(1,4) linkages between N-acetylmuramic acid and N-acetylglucosamine residues in PGN and between N-acetylglucosamine residues in chitooligosaccharides, thus closely resembling metazoan lysozymes (EC 3.2.1.17). Importantly, PGN breakdown products produced by LYS1 are immunogenic in plants, and *LYS1* mutant genotypes were immunocompromised upon bacterial infection. Our findings suggest that plant enzymatic activities, such as LYS1, are capable of generating soluble PRR ligands that might contribute to the activation of immune responses in cells at and surrounding the site of their generation. We also infer that eukaryotic hosts more generally make concerted use of PGN hydrolytic activities and of PRRs in order to cope with bacterial infections.

## Results

### Arabidopsis PGN binding proteins LYM1 and LYM3 are devoid of PGN hydrolytic activity

Soluble oligomeric PGN fragments have previously been shown to stimulate plant immune responses in Arabidopsis (*Gust et al., 2007*; *Erbs et al., 2008*; *Willmann et al., 2011*). As some metazoan PGRPs harbor PGN-degrading enzyme activities (*Gelius et al., 2003*; *Wang et al., 2003*; *Bischoff et al., 2006*; *Dziarski and Gupta, 2010*; *Kurata, 2010*), we tested whether recombinant Arabidopsis PGN binding proteins LYM1 and LYM3 were able to catalyze PGN degradation. For this, we have employed a standard lysozyme assay (*Park et al., 2002*) that is based on reduced turbidity in suspensions of Gram-positive *Micrococcus luteus* cell wall preparations due to PGN degradation. PGN-degrading activity of hen egg-white lysozyme served as a positive control in these assays. As shown in *Figure 1A*, lysozyme, but not recombinant LYM1 or LYM3, displayed cell wall-degrading lytic activity, suggesting that the latter are unable to release PGN fragments from bacterial cell walls. This is in agreement with a lack of sequence similarities between LYM1 or LYM3 and known metazoan PGN hydrolytic activities. We therefore conclude that LYM1 and LYM3 constitute plant PGN sensors that appear to be functionally related to non-enzymatic mammalian or Drosophila PGRPs (*Cho et al., 2005*; *Bischoff et al., 2006*; *Dziarski and Gupta, 2010*; *Kurata, 2010*).

### *LYS1* expression is activated upon bacterial infection

Lysozymes (EC 3.2.1.17) hydrolyze β(1,4) linkages between N-acetylmuramic acid and N-acetylglucosamine residues in PGNs and between N-acetylglucosamine residues in chitodextrins (http://enzyme.expasy.org/EC/3.2.1.17). Plant genomes do not encode lysozyme-like proteins, but many plant species produce lysozyme-like enzyme activities such as chitinases (EC 3.2.1.14) (*Audy et al., 1988*; *Sakthivel et al., 2010*). Plant chitinases fall into five classes (I–V, *Figure 1B*) (*Passarinho and de Vries, 2002*) and are grouped into structurally unrelated families 18 and 19 of glycosyl hydrolases, respectively (*Henrissat, 1991*). Chitinases belonging to family 18 of glycosyl hydrolases are ubiquitously found in all organisms whereas chitinases of glycosyl hydrolase family 19 are found almost exclusively in plants. Class III chitinases (glycosyl hydrolase family 18) represent bifunctional plant enzymes with lysozyme-like activities. One such enzyme, hevamine from the rubber tree *Hevea brasiliensis* (*Beintema et al., 1991*), has been shown to hydrolyze PGN and the structurally closely related β(1,4)-linked GlcNAc homopolymer chitin in vitro (*Bokma et al., 1997*).

To explore host-mediated PGN degradation and its possible implication in plant immune activation, we have addressed the only class III chitinase (which we named LYS1, At5g24090) encoded by the

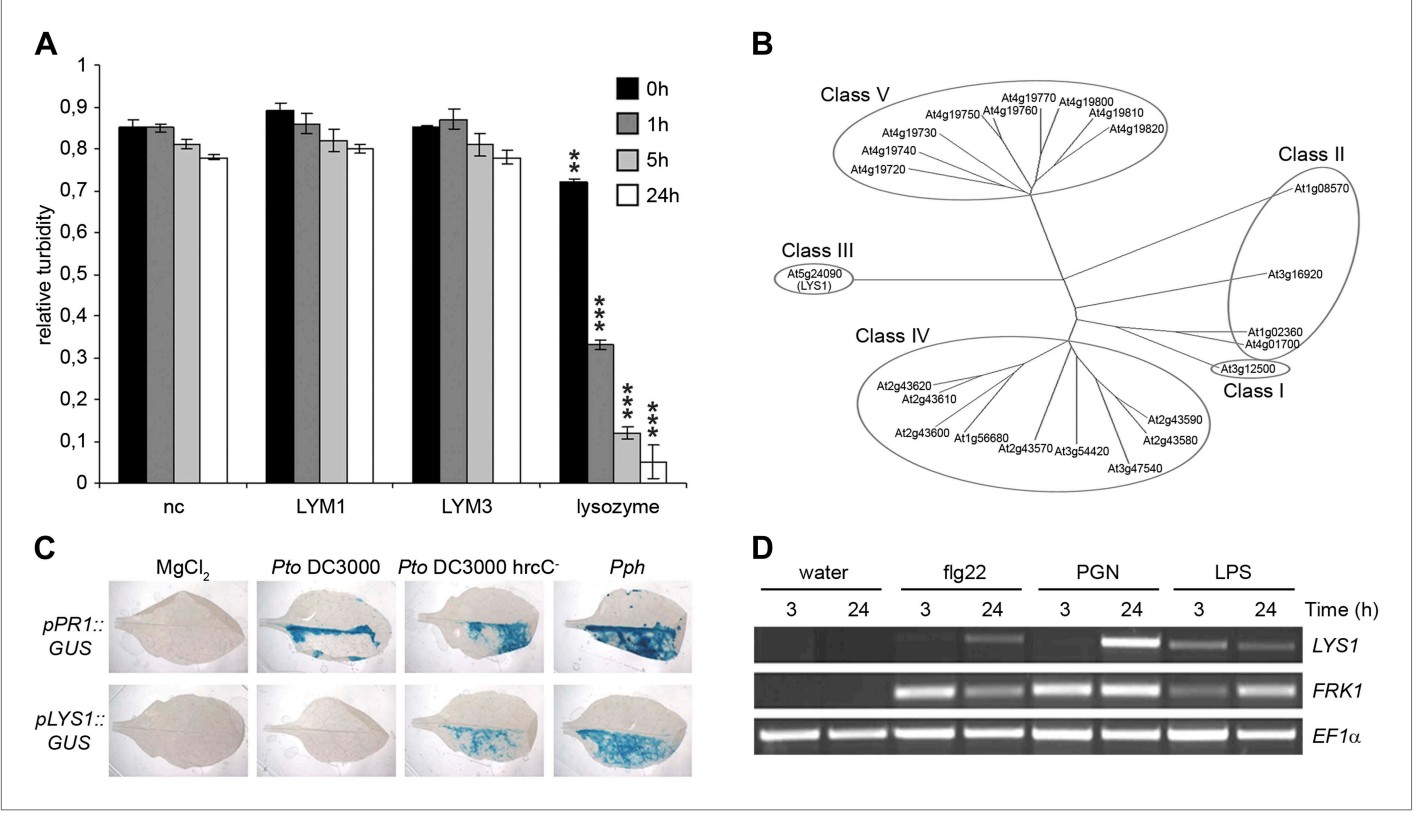

**Figure 1**. The Arabidopsis lysozyme 1 (LYS1) gene is transcriptionally activated upon pathogen-infection. (**A**) LYM1 and LYM3 do not possess peptidoglycan (PGN) hydrolytic activity. *Micrococcus luteus* cell wall preparations were incubated with 20 μg affinity-purified His6-tagged LYM1 or LYM3 or 0.5 μg hen egg-white lysozyme and PGN hydrolytic activity was assayed in a turbidity assay at the indicated time points. As negative control (nc), non-induced His6-tagged LYM3 bacterial lysates were used for affinity purification and eluates were subjected to turbidity assays. Means ± SD of three replicates per sample are given. Statistical significance compared with the negative control (\*\*p<0.001, \*\*\*p<0.0001, Student's *t* test) is indicated by asterisks. (**B**) Multiple sequence alignment of the 24 Arabidopsis chitinases using the ClustalW2 algorithm. Full length amino acid sequences were aligned and subgroups were classified according to **Passarinho and de Vries (2002)**. Arabidopsis lysozyme 1 (LYS1, At5g24090) represents the only member of class III. (**C**) The expression of LYS1 in transgenic pLYS1::GUS reporter plants. Leaf halves of transgenic pLYS1::GUS or pPR1::GUS reporter plants were infiltrated with the virulent *Pseudomonas syringae* pv. *tomato* (*Pto*) DC3000, the type III secretion system-deficient *Pto* DC3000 hrcC⁻ or the avirulent *Pseudomonas syringae* pv. *phaseolicola* (*Pph*) strain (10⁸ cfu/ml) or 10 mM MgCl₂ as control. After 24 hr the leaves were harvested and stained for β-glucuronidase (GUS) activity. (**D**) Leaves of wild-type plants were treated for 3 or 24 hr with 1 μM flg22, 100 μg/ml PGN from *Pto* or 100 μg/ml lipopolysaccharide (LPS). Total RNA was subjected to RT-PCR using *LYS1* or Flagellin-responsive kinase 1 (*FRK1*) specific primers. *EF1α* transcript was used for normalization. All experiments shown in panels (**A**), (**C**) and (**D**) were repeated once with similar results.

Arabidopsis genome (**Passarinho and de Vries, 2002**; **Figure 1B**). Bacterial infection of Arabidopsis plants stably expressing a *pLYS1::GUS* construct revealed that *LYS1* gene expression is enhanced upon infection with host non-adapted *Pseudomonas syringae* pv. *phaseolicola* (*Pph*) or disarmed host adapted *P. syringae* pv. *tomato* (*Pto*) DC3000 hrcC⁻. Likewise, expression of the immune response marker *pathogenesis-related protein 1* (*PR1*) was enhanced by the same treatment (**Figure 1C**). Failure to detect *LYS1* expression in plants infected with virulent host adapted *Pto* DC3000 suggests bacterial effector-mediated suppression that is reminiscent of that observed for PGN receptor proteins LYM1 and LYM3 (**Willmann et al., 2011**) as well as numerous other immunity-associated genes (**Kemmerling et al., 2007**; **Postel et al., 2010**). *LYS1* gene expression is not only triggered upon bacterial infection, but was also observed upon treatment with different MAMPs including bacterial flagellin, LPS, or PGN preparations (**Figure 1D**), similar to the immune marker gene *Flagellin-responsive kinase 1* (*FRK1*). Altogether, infection-induced *LYS1* transcriptional activation suggests that the LYS1 protein is implicated in immunity to bacterial infection.

## LYS1 is a plant lysozyme

To analyze the enzymatic properties of LYS1, recombinant protein production was attempted. Overexpression in *Escherichia coli* failed to produce active enzyme and LYS1 production in eukaryotic

*Pichia pastoris* entirely failed to produce recombinant protein (not shown). Therefore, we resorted to generate *p35S::LYS1-GFP*-overexpressing (*LYS1^{OE}*) plants (**Figure 2A,B**). Notably, LYS1-GFP was glycosylated (**Figure 2C**), possibly explaining the failure to produce enzymatically active LYS1 protein in *E. coli*. Expression of the green fluorescent protein (GFP) fusion protein in Arabidopsis plants was accompanied by substantial proteolytic cleavage resulting in the predominant release of a protein with an approximate molecular mass of 35 kDa, most likely representing untagged LYS1 (**Figure 2B**). Analysis of this major cleavage product by liquid chromatography-mass spectrometry (LC-MS/MS) after tryptic in-gel digestion and by peptide mass fingerprint not only confirmed the identity of LYS1 in this band but also yielded peptides spanning the whole protein sequence, except for the first 53 amino acids (data not shown), thus indicating cleavage of the LYS1-GFP fusion protein between LYS1 and GFP.

Three mutant lines with T-DNA insertions in the LYS1 gene were available from the Nottingham Arabidopsis Stock Centre. However, neither the insertion in the 5' untranslated region nor the insertions in the first intron and at the end of the last exon of the coding region abolished formation of the *LYS1* transcript (**Figure 3—figure supplement 1**). As an alternative to knock-out lines, *LYS1* knock-down lines (*LYS1^{KD}*) were produced by artificial micro RNA technology (**Schwab et al., 2006**; **Figure 3**). As proven by quantitative reverse transcriptase polymerase chain reaction (RT-qPCR), we obtained two genetically independent *LYS1^{KD}* lines with residual transcript levels not exceeding 10% of those detected in wild-type plants (**Figure 3C**). In contrast, the transcription of potential off-target genes was not affected (**Figure 3C**). Protein extracts derived from transgenic plants were tested for chitinolytic activity by employing 4-methylumbelliferyl β-D-N, N', N"-triacetylchitotriose (4-MUCT) as substrate. Leaf protein extracts from *LYS1^{OE}* plants exhibited significant chitinase activity when compared with a *Streptomyces griseus* chitinase control (**Figure 4A**). In contrast, wild-type and *LYS1^{KD}* plants exhibited only marginal chitinase activities. Likewise, using 4-MUCT in a gel electrophoretic separation-based chitinase assay produced a zymogram in which enzyme activity was solely detectable in protein extracts obtained from *LYS1^{OE}* plants, but not in those from control plants expressing secreted GFP (*secGFP*) (**Figure 4B**). Thus, LYS1 indeed harbors the predicted chitinase activity. As 4-MUCT is also a typical substrate for lysozymes (**Brunner et al., 1998**), this was the first indication that LYS1 might also harbor lysozyme activity. Next, leaf protein extracts from *LYS1^{OE}* plants were tested for their ability to solubilize complex PGN presented by intact Gram-positive *M. luteus* cells and to cleave preparations of complex, insoluble *Bacillus subtilis* PGN. Again, protein extracts from *LYS1^{OE}* plants exhibited significant PGN-degrading activity whereas wild-type and *LYS1^{KD}* plants showed basal activity levels only (**Figure 4C,D**). Likewise, PGN-solubilizing activity profiles of protoplast suspensions derived from these transgenics confirmed significant PGN-degrading activity of *LYS1^{OE}* plants (**Figure 4E**).

To determine specific enzyme activities, untagged LYS1 was purified from *LYS1^{OE}* Arabidopsis lines by fast protein liquid chromatography (FPLC) and used for enzyme assays. The 4-MUCT assay yielded a Michaelis constant ($K_m$) of 70 ± 14 µM and a $V_{max}$ of 378 ± 42 µM min$^{-1}$ mg$^{-1}$ for LYS1, and a $K_m$ of 53 ± 27 µM and a $V_{max}$ of 397 ± 145 µM min$^{-1}$ mg$^{-1}$ for commercial *S. griseus* chitinase. Using the turbidity assay with *M. luteus* cell wall preparations, a $K_m$ of 18.2 ± 2.5 mg/ml and $V_{max}$ of 4.4 ± 0.6 mg mg$^{-1}$ min$^{-1}$ were obtained for LYS1, and a $K_m$ of 8.4 ± 0.8 mg/ml and $V_{max}$ of 192 ± 120 mg mg$^{-1}$ min$^{-1}$ for commercial hen egg-white lysozyme. The $K_m$ values for LYS1 are thus comparable to the commercial enzymes.

As shown in **Figure 4E**, the majority of LYS1 activity was found in the supernatant of the protoplasts, suggesting an apoplastic localization of LYS1. To confirm this localization we prepared apoplastic washes from *LYS1^{OE}* Arabidopsis lines. Both the LYS1-GFP fusion protein as well as free LYS1 was detectable in concentrated apoplastic fluids whereas the cytoplasmic mitogen-activated protein kinase MPK3 was only present in the total leaf protein samples (**Figure 4—figure supplement 1A**). Moreover, transient expression in the heterologous plant system *Nicotiana benthamiana* of the *p35S::LYS1-GFP* construct resulted in labeling of the cell periphery, whereas expression of a construct lacking the *LYS1* signal peptide-encoding sequence yielded labeling of intracellular structures (**Figure 4—figure supplement 1B**). Use of the fluorescent dye FM4-64, a plasma membrane and early endosome marker (**Bolte et al., 2004**), revealed that LYS1 signals co-localized to a large extent with the plasma membrane (**Figure 4—figure supplement 1B**). Thus, LYS1 likely operates in close vicinity of the plant surface. Indeed, previous identification within the Arabidopsis cell wall proteome (**Kwon et al., 2005**) suggests that LYS1 acts in the plant apoplast. Since the plant apoplast is an acidic compartment (pH 5–6) (**Schulte et al., 2006**), we investigated whether LYS1 is active at physiologically relevant pH conditions. For this, the *M. luteus* cell wall-degrading activity of an *LYS1^{OE}* leaf extract was

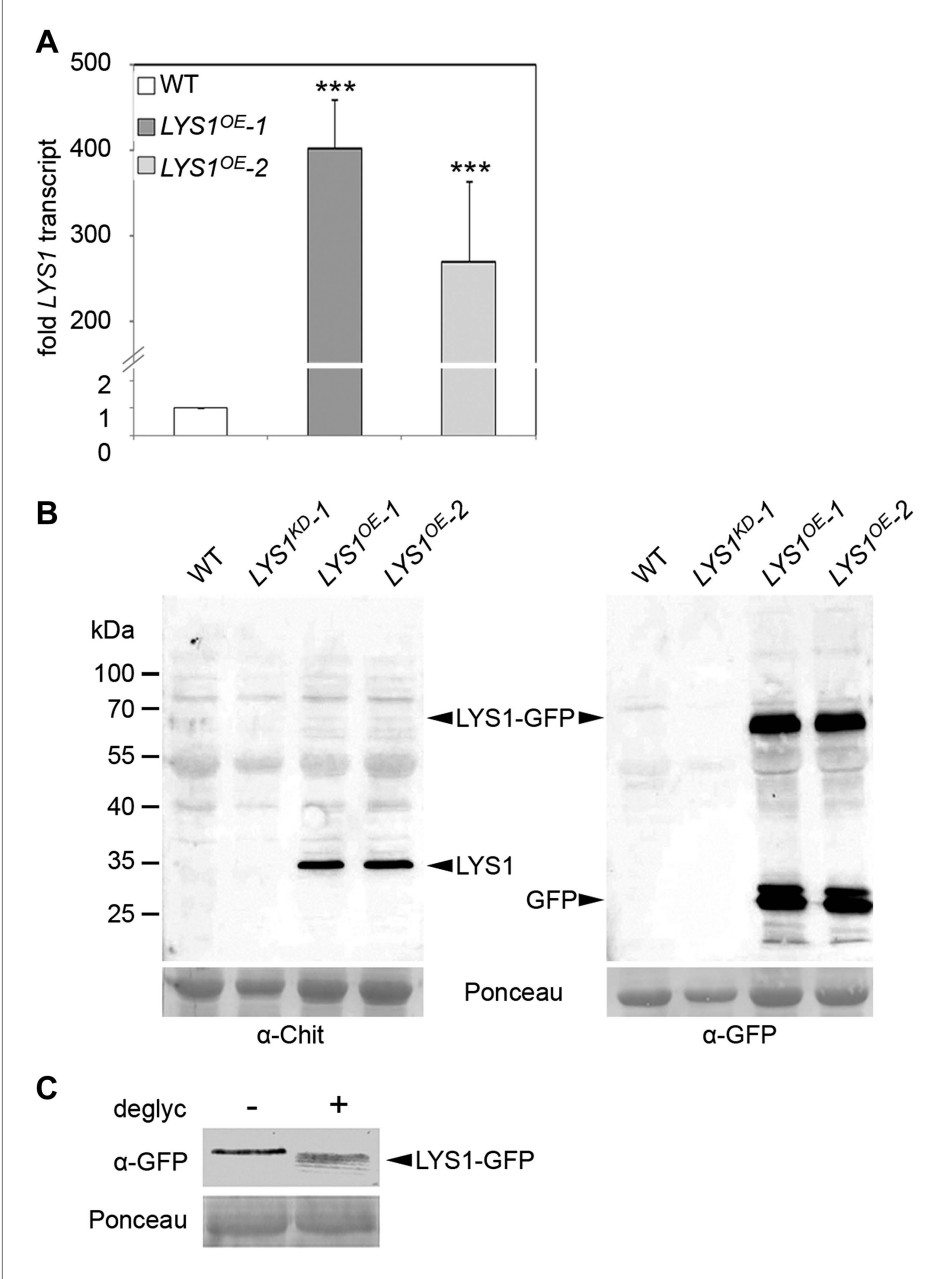

**Figure 2**. Analysis of LYS1 overexpression lines. (**A**) RT-qPCR analyses of transcript levels in mature leaves of two independent transgenic lines expressing *p35S::LYS1-GFP* (*LYS1^OE^-1*, *LYS1^OE^-2*) relative to expression levels in wild-type. *EF1α* transcript was used for normalization. Error bars, SD (n = 3). Statistical significance compared with wild-type (***p<0.001, Student's *t* test) is indicated by asterisks. (**B**) Immunoblot analysis of protein extracts from leaves of two independent *LYS1^OE^* lines, a LYS1 knock-down line (*LYS1^KD^-1*, see *Figure 3*) and wild-type plants. Total leaf protein was separated by SDS-PAGE and blotted onto a nitrocellulose membrane. Immunodetection was carried out using α-tobacco class III chitinase (α-Chit) or green fluorescent protein (α-GFP) (both from rabbit) and an anti-rabbit HRP-coupled secondary antibody. Ponceau S red staining of the large subunit of RuBisCO served as loading control. (**C**) Total protein extracts from leaves of *LYS1^OE^-1* plants were subjected to deglycosylation with a deglycosylation kit (NEB). The negative control (−) was treated as the deglycosylation sample (+) but without addition of the deglycosylation enzyme mix. Immunoblot analysis was carried out as described in (**B**). All experiments shown were repeated at least once.

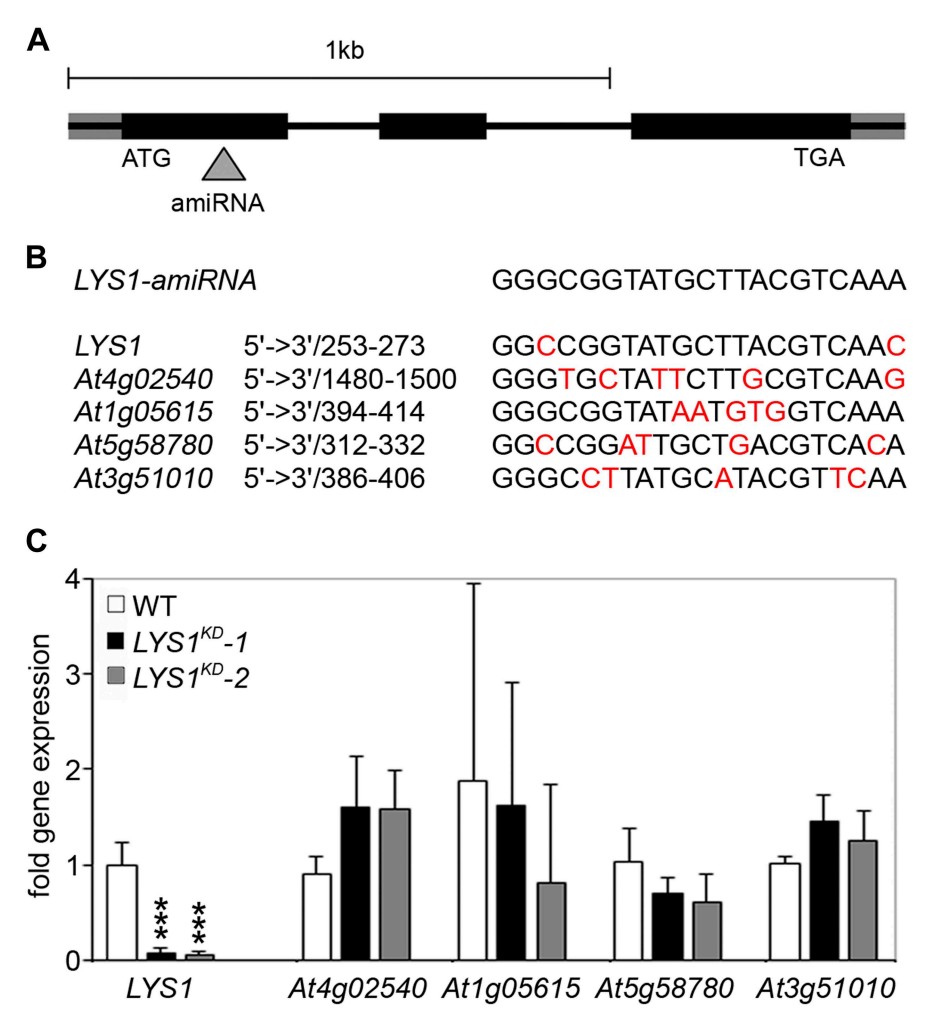

**Figure 3**. Analysis of *LYS1* amiRNA lines. (**A**) Predicted *LYS1* gene structure (exons, black bars; introns, black lines; untranslated regions, gray). The region targeted by the amiRNA construct is indicated by an arrowhead. (**B**) Off-target genes for the *LYS1-amiRNA* construct were identified using the Web microRNA Designer (WMD; http://wmd. weigelworld.org). The region targeted by the amiRNA is given for each gene, mismatches are indicated in red. Potential off targets either possess more than one mismatch at positions 2–12 or have mismatches at position 10 and/or 11 which will limit amiRNA function. (**C**) Transcript levels of the four top hits shown in (**B**) were determined by RT-qPCR in untreated seedlings of two independent transgenic *LYS1*-amiRNA knock-down lines (*LYS1^{KD}-1*, *LYS1^{KD}-2*) using gene-specific primers for *LYS1* (*At5g24090*), *At4g02540*, *At1g05615*, *At5g58780*, and *At3g51010*. *EF1α* transcript was used for normalization. Error bars, SD (n = 3). Statistical significance compared with the wild-type control (which was set to 1 for each primer set) is indicated by asterisks (\*\*\*p<0.001, Student's *t* test). The experiment was repeated once with similar results.

The following figure supplements are available for figure 3:

**Figure supplement 1**. Characterization of LYS1 T-DNA insertion lines.

determined at different pH values. Although active at pH values ranging from 3.2 to 7.2, a pronounced maximum of LYS1 activity was detected around pH 6 which coincided with the apoplastic pH of plant cells (*Figure 4F*).

To further confirm LYS1 glucan hydrolytic activity, an epitope-tagged *LYS1* fusion construct was transiently expressed in *N. benthamiana* (*Figure 5A*). Similar to the Arabidopsis *LYS1^{OE}* leaf extracts, extracts from *p35S::LYS1-myc* expressing *N. benthamiana* leaves also displayed in-gel chitinolytic activity (*Figure 5B*) compared with extracts from control leaves expressing the viral silencing suppressor

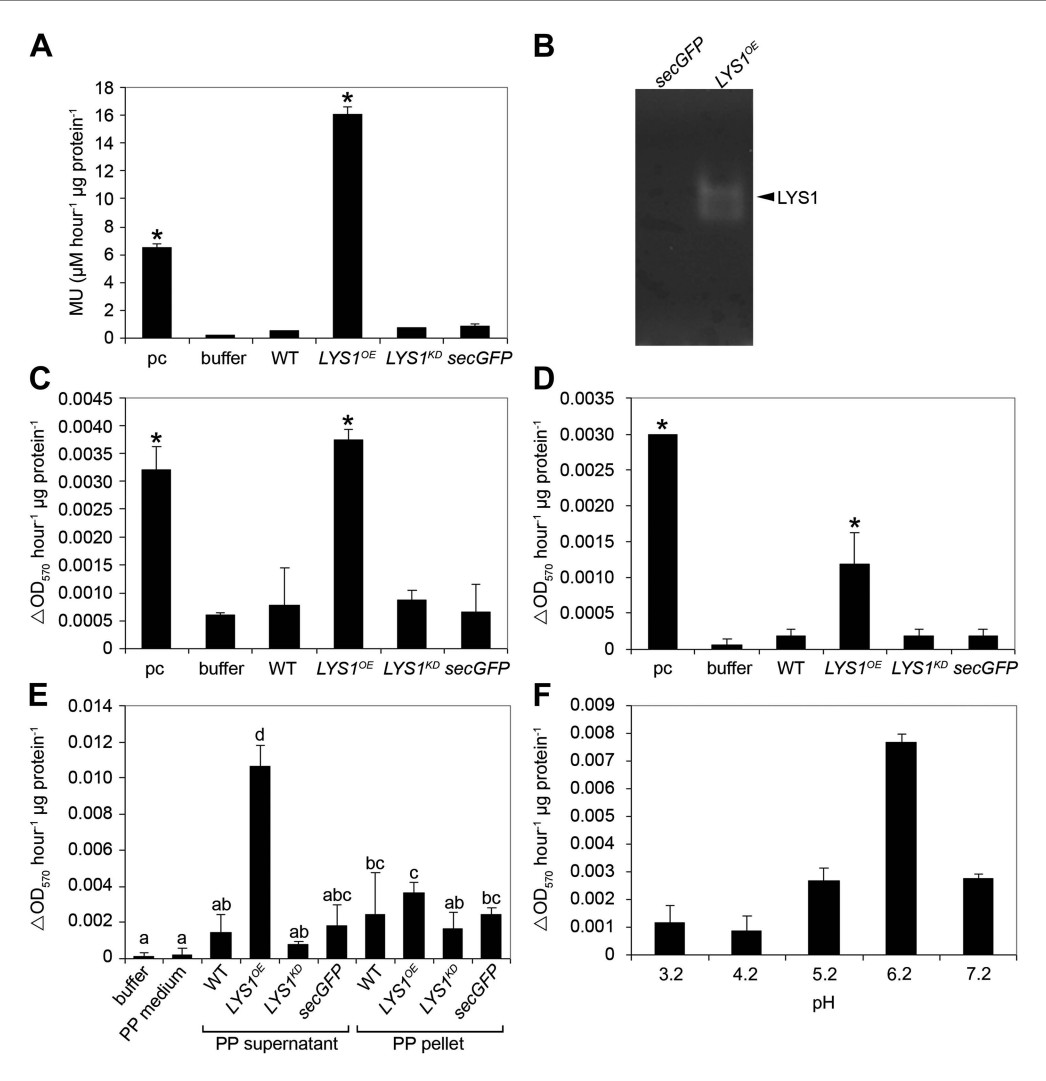

**Figure 4**. LYS1 is a glucan hydrolase. (**A-D**) Protein extracts from adult wild-type or *LYS^OE^-1* and *LYS^KD^-1* homozygous lines were assayed for hydrolytic activity towards glycan substrates. Plants expressing secreted green fluorescent protein (GFP) (*secGFP*) served to control the effect of external GFP. (**A**) Leaf protein extracts from indicated transgenic plants were assayed for chitinolytic activity using the 4-methylumbelliferyl β-D-N, N', N"-triacetylchitotriose (4-MUCT) substrate. Enzymatic activities 4 hr after treatment were calculated using *Streptomyces griseus* chitinase as positive control (pc). (**B**) Protein extracts from *LYS1^OE^-1* or *secGFP* plants were separated on a cetyltrimethylammonium bromide-polyacrylamide gel and hydrolytic activity was assayed by overlaying the gel with the substrate 4-MUCT. Fluorescent bands are indicative of substrate cleavage. The arrowhead indicates the position of LYS1. (**C** and **D**) *Micrococcus luteus* cells (**C**) or *Bacillus subtilis* peptidoglycan (PGN) (**D**) were subjected to hydrolysis by leaf protein extracts and PGN hydrolytic activity was calculated after 4 hr using hen egg-white lysozyme as positive control (pc). Significant differences compared with the buffer control are indicated by asterisks (*p<0.05; Student's *t* test; **A**, **C**, **D**). (**E**) Protoplasts of transgenic lines were pelleted and protein extracts of the protoplast (PP) pellet or medium supernatant were subjected to the PGN hydrolysis assay as described in (**C**). As controls, buffer or protoplast medium (PP medium) was used. Means ± SD of two replicates per sample are given, bars with different letters are significantly different based on one-way ANOVA (p<0.05). (**F**) Lysis of *M. luteus* cells was determined in a turbidity assay with *LYS1^OE^* leaf protein extracts as described in (**C**) at the indicated pH. Means ± SD of two replicates per sample are given. All experiments shown were repeated at least once.

The following figure supplements are available for figure 4:

**Figure supplement 1**. LYS1 is located in the plant apoplast.

**Figure supplement 2**. LYS1 is devoid of cellulose hydrolytic activity.

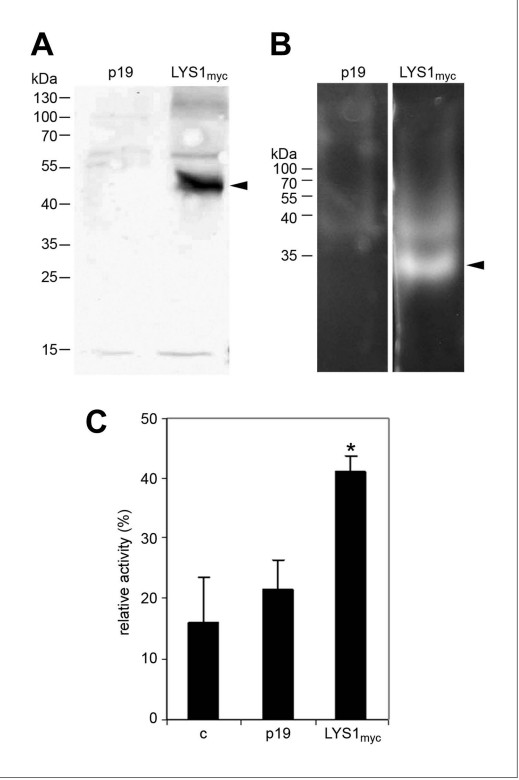

**Figure 5**. LYS1 transiently expressed in *Nicotiana benthamiana* possesses hydrolytic activity. (**A**) Protein extracts from *N. benthamiana* leaves expressing LYS1 fused to the myc-epitope tag under control of the *p35S* promoter were separated on an SDS-polyacrylamide gel and analyzed by western blot using antibodies raised against the myc-epitope tag. As control, plants were infiltrated with agrobacteria harboring the p19 suppressor of silencing construct (p19). Protein sizes (kDa) are indicated on the left. (**B**) *N. benthamiana* protein extracts from leaves expressing *LYS1myc* or p19 were separated on a cetyltrimethylammonium bromide-polyacrylamide gel and hydrolytic activity was assayed by overlaying the gel with the substrate 4-methylumbelliferyl β-D-N, N', N''-triacetylchitotriose. Fluorescent bands are indicative of substrate cleavage. Arrowheads indicate the positions of epitope-tagged LYS1. (**C**) Protein extracts from *N. benthamiana* leaves expressing *LYS1myc* or p19 were assayed for peptidoglycan (PGN) hydrolytic activity in a turbidity assay using *Bacillus subtilis* PGN. Relative activities (2 hr post treatment) were calculated using hen egg-white lysozyme as standard. Statistical significance compared with the untreated control (*p<0.05, Student's *t* test) is indicated by asterisks. All experiments shown were repeated at least once.

p19 only. Likewise, *N. benthamiana* protein extracts containing LYS1-myc were able to cleave preparations of complex insoluble *B. subtilis* PGN (**Figure 5C**).

In sum, we provide biochemical evidence that LYS1 harbors hydrolytic activity for chitin as well as for PGN of the lysine-type (*M. luteus*) and diaminopimelic acid-type (*B. subtilis*). Importantly, LYS1 failed to exhibit activity on cellobiose as a substrate, indicating it might have no cellulose activity (**Figure 4—figure supplement 2**). Thus, LYS1 resembles enzymatic activities reported for metazoan lysozymes and should be classified as lysozyme (EC 3.2.1.17) instead of chitinase (EC 3.2.1.14).

## LYS1 generates plant immunogenic PGN fragments

To analyze immunogenic activities of PGN cleavage products generated by LYS1, untagged LYS1 was purified from *LYS1^OE* Arabidopsis lines by FPLC and used for degradation of *B. subtilis* PGN. Solubilized PGN fragments found in the supernatant of LYS1-digested PGN were subsequently analyzed by high performance liquid chromatography (HPLC) (**Figure 6A**). Only a few peaks could be detected in the supernatant of PGN incubated with a buffer control or with heat-inactivated LYS1. In contrast, PGN digests produced by native LYS1 yielded several characteristic peaks that were also detectable in the supernatants of PGN preparations treated with mutanolysin, which has been shown to cleave O-glycosidic bonds between GlcNAc and MurNAc residues in complex PGN (**Yokogawa et al., 1975**). LYS1-generated PGN fragments were subsequently tested for their ability to trigger plant immunity-associated responses (**Figure 6B–D**). First, supernatants of PGN preparations treated with either native or heat-denatured LYS1 were used to trigger immune marker gene *FRK1* expression in Arabidopsis seedlings. Importantly, only supernatants from PGN digests produced by native LYS1 or mutanolysin induced *FRK1* expression whereas buffer controls or digests produced by heat-inactivated LYS1 did not release immunogenic soluble fragments from complex PGNs (**Figure 6B**). Notably, activation of immune responses by LYS1-generated PGN fragments was dependent on Arabidopsis PGN receptor complex components LYM1, LYM3, and CERK1 as the respective mutant genotypes failed to respond to immunogenic PGN fragments

(**Figure 6B**). Second, we tested whether LYS1-generated PGN fragments were able to trigger an immunity-associated response, medium alkalinization, in rice cell suspensions. This plant was chosen for testing as a PGN receptor system very similar to that in Arabidopsis has recently been reported (**Liu et al., 2012a**). As shown in **Figure 6C**, LYS1-released PGN fragments triggered medium alkalinization in cultured

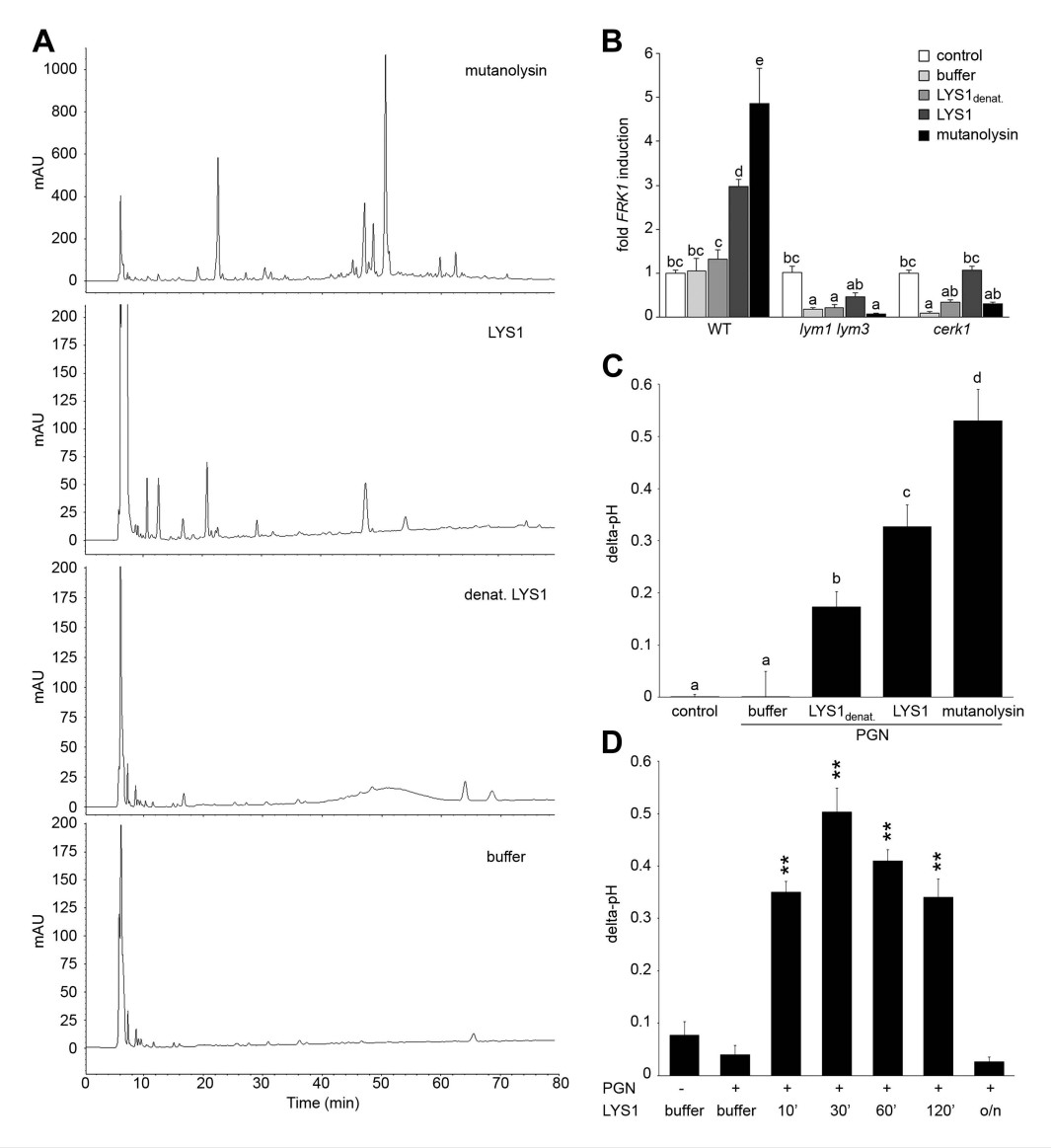

**Figure 6**. Purified LYS1 generates immunogenic peptidoglycan (PGN) fragments. LYS1 was purified from 5-week-old *LYS1^OE* plants and used for PGN digestion. (**A**) 500 µg *Bacillus subtilis* PGN were digested for 7 hr with mutanolysin (50 µg/ml), native purified LYS1 (140 µg/ml), heat-denatured purified LYS1 (140 µg/ml), or the reaction buffer alone and subjected to high performance liquid chromatography fractionation. Shown are the peak profiles of representative runs. The signal intensity is given in milliabsorbance units (mAU). (**B**) *B. subtilis* PGN was digested for 4 hr as described in (**A**) and Arabidopsis wild-type seedlings or the indicated mutant lines were treated for 6 hr with 25 µl/ml digest supernatant containing solubilized PGN fragments. Total seedling RNA was subjected to RT-qPCR using *Flagellin responsive kinase* (*FRK1*) specific primers. *EF1α* transcript was used for normalization, water treatment served as control and was set to 1. (**C**) Supernatants of digested PGN (25 µl/ml) were added to cultured rice cells and medium alkalinization was determined 20 min post addition. Treatment with water or MES buffer served as control. All data represent triplicate samples ± SD, bars with different letters are significantly different based on one-way ANOVA ($p<0.05$; **B** and **C**). (**D**) *B. subtilis* PGN was digested with native purified LYS1 for the indicated times or overnight (o/n) and digest supernatant was used to trigger medium alkalinization in rice cells as described in (**C**). All data represent triplicate samples ± SD, asterisks indicate significant differences compared to the buffer control (*$p<0.05$; **$p<0.01$; ***$p<0.001$; Student's *t* test). All experiments shown were repeated at least once.

rice cells, suggesting that immune defense stimulation by soluble PGN fragments is not restricted to Arabidopsis only.

We further investigated the kinetics of PGN fragment release from complex PGNs. As shown in *Figure 6D*, release of immunogenic PGN fragments into solution occurred rapidly within 10 min of incubation with native LYS1. Incubation of complex PGNs with LYS1 yielded the highest immunogenic activity of the digest supernatant after 30 min, suggesting that at that time point the maximum amount of immunogenic PGN fragments was generated. However, prolonged incubation with LYS1 again resulted in a loss of activity with overnight digestion completely abolishing stimulatory activity of the PGN digest. We assume that LYS1 is capable of releasing immunogenic fragments from complex PGNs, but extensive or complete digest into PGN monomers or small PGN fragments appears to abolish the immunogenic activity of PGN fragments. This result is in accordance with our previous observations that prolonged digestion of PGN with mutanolysin diminishes its defense-inducing activity (*Gust et al., 2007*).

## LYS1 is required for plant immunity towards bacterial infections

To examine the physiological role of LYS1 in plant immunity, *LYS1^OE* and *LYS1^KD* lines were subjected to infection with various phytopathogens. As LYS1 harbors chitinase activity (*Figures 4A,B and 5B*) and as *LYS1* transcripts accumulate upon fungal infection (*Samac and Shah, 1991*), we first analyzed the role of LYS1 in immunity towards fungal infection. Leaves of transgenic *LYS1^OE* or *LYS1^KD* lines and wild-type plants were infected with the necrotrophic fungus *Botrytis cinerea* and disease symptoms were monitored 2–3 days post infection. Fungal hyphal growth and necrotic leaf lesions at infection sites were detectable in all plant lines tested and hyphal outgrowth or cell death lesion sizes revealed no differences between wild-type, *LYS1^OE* or *LYS1^KD* lines (*Figure 7*). Likewise, infection with the necrotrophic fungus *Alternaria brassicicola* resulted in indistinguishable necrotic lesions in *LYS1^OE* and *LYS1^KD* transgenics compared to those observed in wild-type control plants (*Figure 8*). Trypan blue staining and microscopic analysis of the infection sites did not reveal major differences in fungal hyphal growth among all lines tested (*Figure 8B,C*). Although disease indices at day 11 after infection were slightly increased in *LYS1^KD* lines (*Figure 8D*), such subtle differences were not statistically significant. In conclusion, we failed to detect a role for LYS1 in immunity to fungal infection with *B. cinerea* and *A. brassicicola* under our experimental conditions. However, these results cannot be generalized and LYS1 might still have a role under infection regimes other than the ones used here or it might be important for defense against other fungal pathogens.

To examine the role of LYS1 in immunity to bacterial infection, we infected wild-type plants or *LYS1^KD* and *LYS1^OE* lines with virulent *Pto* DC3000. Two independent *LYS1^KD* lines exhibited hypersusceptibility to bacterial infection (*Figure 9A*), suggesting that lack of PGN-degrading activity results in reduced plant immunity. Likewise, immunity to hypovirulent *Pto* DC3000 Δ*AvrPto/PtoB* was compromised in these lines (*Figure 9B*). Moreover, expression of the immune marker gene *FRK1* upon administration of complex PGNs was greatly impaired in the *LYS1^KD* mutants (*Figure 9C*). These findings suggest that the enzymatic activity of LYS1 on PGN contributes substantially to plant immunity against bacterial infection.

Unexpectedly, bacterial growth on *LYS1^OE* lines was also significantly enhanced compared with that observed on wild-type plants (*Figure 9A,B*). *FRK1* transcript accumulation upon administration of complex PGN was also strongly reduced in *LYS1* overexpressors (*Figure 9C*). To exclude a direct effect of LYS1 overexpression on PGN receptor abundance, we examined transcript levels of *LYM1*, *LYM3*, and *CERK1* but found no effect on the transcription of these receptor genes in the *LYS1^OE* lines (*Figure 9—figure supplement 1A*). Also, CERK1 protein levels were unaltered in the *LYS1^OE* lines, whereas there was no CERK1 protein detectable in the *cerk1-2* mutant (*Figure 9—figure supplement 1A*). Moreover, we included the *LYS1^OE*-3 line with only moderately increased *LYS1* transcript and protein levels in mature leaves (*Figure 9—figure supplement 1A,B*). Susceptibility to *Pseudomonas* infection in the *LYS1^OE*-3 line was only slightly but not significantly increased (p=0.064, Student's *t* test). These results indicate that lowering *LYS1* expression levels, accompanied by lower LYS1 hydrolytic activity on PGN, brings down these lines close to wild-type. Thus, massive *LYS1* overexpression and loss-of-function mutations are phenocopies of each other, irrespective of the fact that *LYS1^KD* and *LYS1^OE* lines show dramatic differences in LYS1 enzymatic activities (*Figure 4*).

Altogether, we propose that LYS1 contributes to plant immunity to bacterial infection by decomposition of bacterial PGNs and generation of soluble PGN-derived patterns that trigger immune activation in a LYM1-LYM3-CERK1 receptor-complex-dependent manner.

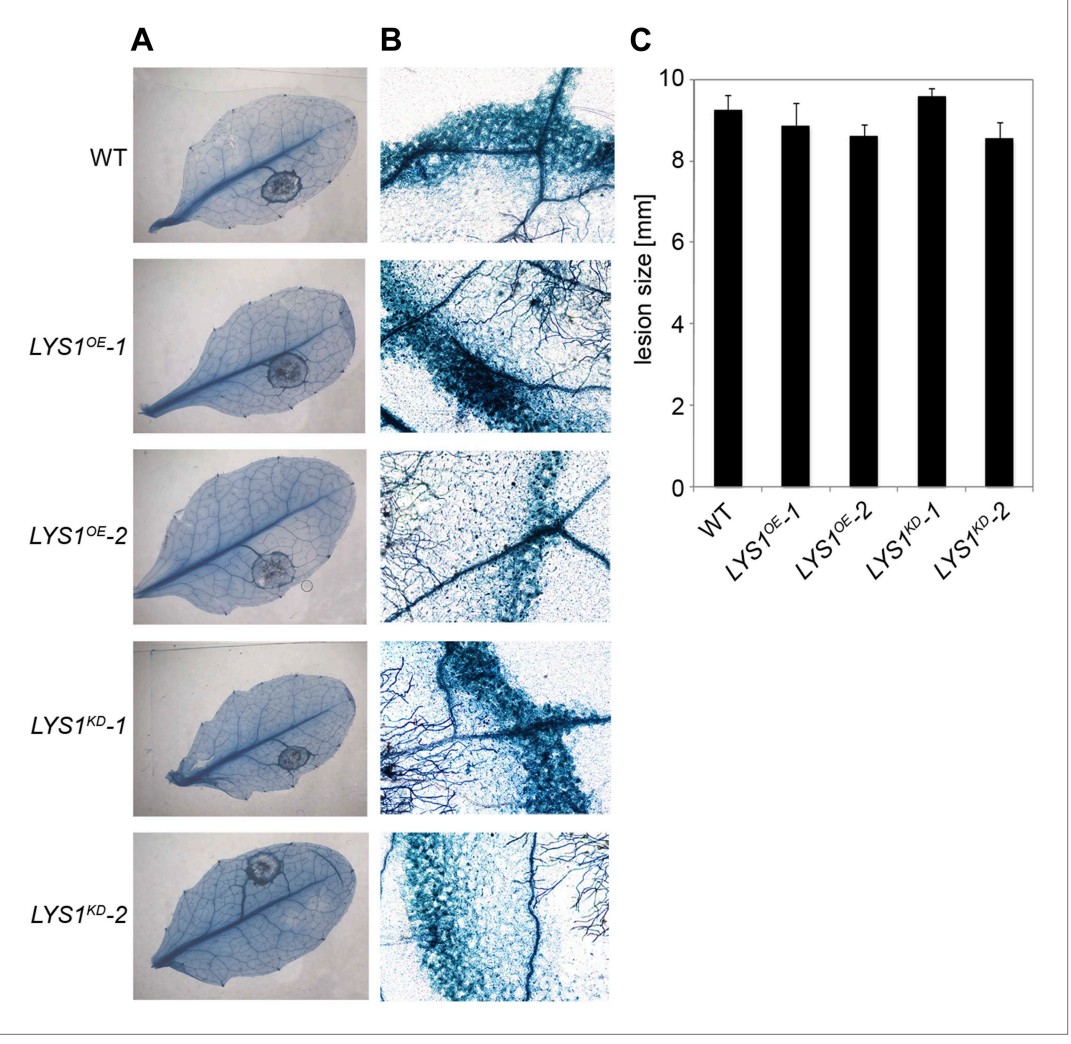

**Figure 7**. *LYS1* lines are not impaired in resistance towards infection with *Botrytis cinerea*. Five-week-old plants were infected with the necrotrophic fungus *Botrytis cinerea*. 5 µl spore suspension of $5 \times 10^5$ spores/ml was drop-inoculated on one half of the leaf; two leaves per plant were infected. The plants were analyzed for development of symptoms 2 and 3 days post infection (dpi). (**A**) Trypan blue stain showing visible symptoms after 2 dpi. (**B**) Microscopic analysis of the infection site and fungal hyphae 2 dpi visualized by Trypan blue stain. (**C**) Measurement of lesion size 3 dpi. Shown are means and standard errors (n = 16). No significant differences were observed (Student's *t* test). The experiment was repeated once with the same result.

## Discussion

It is generally little understood whether and how microbial patterns derived from complex extracellular assemblies, such as bacterial cell walls, are accessible to host PRRs for host immune activation in eukaryotes. This holds true for bacterial PGNs, but also for other patterns including bacterial LPS, flagellin, or fungus-derived chitin or glucan structures, all of which have been ascribed triggers of innate immunity in metazoans and plants (*Boller and Felix, 2009*; *Kumar et al., 2013*; *Newman et al., 2013*; *Pel and Pieterse, 2013*). Limited insight into the 3D structure of ligand–PRR complexes, as well as knowledge on ligand structural requirements for plant immune activation, suggests that small ligand epitopes are crucial for binding to host PRRs (*Liu et al., 2012b*; *Sun et al., 2013*). It is thus generally assumed that soluble fragments derived from complex microbial matrices serve as ligands for host PRRs and subsequent immune activation in both lineages.

Two possible scenarios as to how soluble PGN fragments might be generated from macromolecular assemblies of cross-linked PGNs are discussed. First, during bacterial multiplication and cell wall

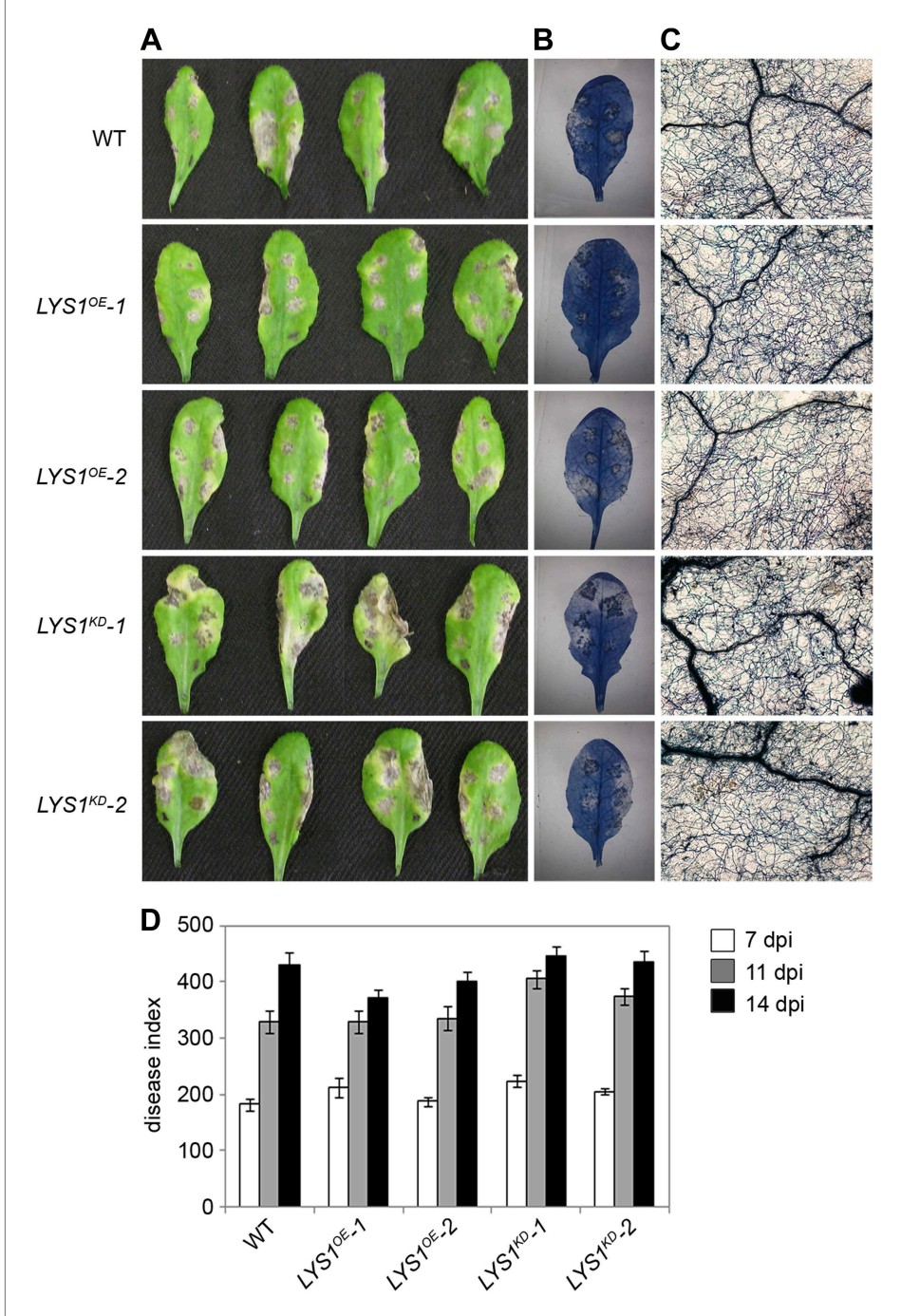

**Figure 8**. *LYS1* mutation does not impinge on resistance towards *Alternaria brassicicola*. Five-week-old plants were infected with the necrotrophic fungus *Alternaria brassicicola*. Six 5 µl droplets of a spore suspension of $5 \times 10^5$ spores/ml were inoculated on the leaf; two leaves per plant were infected. The plants were analyzed for symptom development 7, 11, and 14 days post infection (dpi). (**A**) Visible symptoms of four independent leaves at 14 dpi. (**B**) Disease symptoms 14 dpi visualized by Trypan blue stain. (**C**) Microscopic analysis of the infection site and fungal hyphae 14 dpi visualized by Trypan blue stain. (**D**) Calculation of the disease index at 7, 11, and 14 dpi. Shown are means and standard errors (n = 16). No significant differences were observed (Student's *t* test). The experiment was repeated once with the same result.

biogenesis, large portions of soluble PGN fragments are shed into the extracytoplasmic space from which only 50–90% are recycled (*Park and Uehara, 2008*; *Reith and Mayer, 2011*; *Johnson et al., 2013*). This implies that imperfect recycling of bacterial walls might serve as a source of soluble ligands

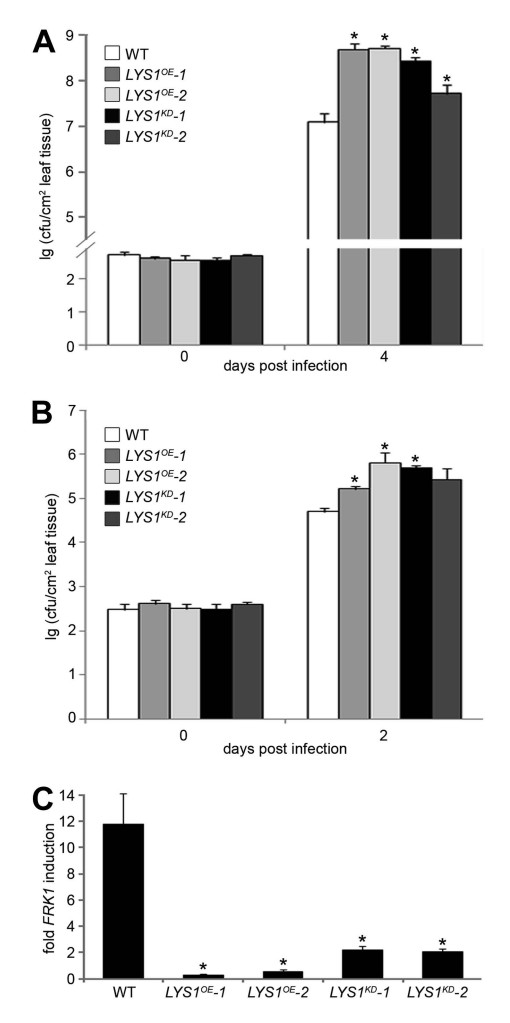

**Figure 9**. Manipulation of *LYS1* levels causes hyper-susceptibility towards bacterial infection and loss of peptidoglycan (PGN)-triggered immune responses. (**A** and **B**) Transgenic *LYS1* plants are hypersusceptible to bacterial infection. Growth of *Pseudomonas syringae* pv. *tomato* (*Pto*) DC3000 (**A**) or *Pto* DC3000 Δ*AvrPto/AvrPtoB* (**B**) was determined 2 or 4 days post infiltration of $10^4$ colony forming units ml$^{-1}$ (cfu/ml). Data represent means ± SD of six replicate measurements/genotype/data point. Representative data of at least four independent experiments are shown. (**C**) Transgenic *LYS1* plants are impaired in PGN-induced immune gene expression. Leaves of wild-type plants or transgenic *LYS1* plants were treated for 6 hr with 100 µg *Bacillus subtilis* PGN and total RNA was subjected to RT-qPCR using *Flagellin responsive kinase* (*FRK1*) specific primers. *EF1α* transcript was used for normalization. Data represent means ± SD of triplicate samples, and shown is the result of one of three independent experiments. Statistical significance compared with wild-type (*p<0.05, Student's *t* test) is indicated by asterisks.

*Figure 9. Continued on next page*

for host PRRs sensing PGN (*Boudreau et al., 2012*; *Wyckoff et al., 2012*). Indeed, muramyl-peptides spontaneously shed by *Shigella flexneri* directly stimulate NOD1-dependent immune responses in mammalian immune cells, and bacterial mutants impaired in PGN recycling hyperactivate host immunity (*Nigro et al., 2008*). Second, host lysozyme activity has been demonstrated to generate soluble PGN ligands for NOD2 receptor-mediated immune activation and clearance of *Streptococcus pneumoniae* colonization in mice (*Callewaert and Michiels, 2010*; *Clarke and Weiser, 2011*; *Davis et al., 2011*). Importantly, *Davis et al. (2011)* established a role for host lysozymes in PGN release from bacteria in the absence of detectable bacterial lysis. Likewise, Drosophila Gram-negative bacteria-derived binding protein 1 (GNBP1) was shown to possess PGN-hydrolyzing activity and to deliver fragmented PGN to the PGN sensor, PGRP-SA (*Filipe et al., 2005*; *Wang et al., 2006*). Thus, both passive and active mechanisms of PGN decomposition appear to occur simultaneously during host pathogen encounters and might not be mutually exclusive.

We here report on a lysozyme-like enzyme (LYS1) that is produced in infected Arabidopsis plants and is capable of generating soluble PGN fragments from complex bacterial PGNs. LYS1 has been demonstrated to hydrolyze β(1,4) linkages between N-acetylmuramic acid and N-acetylglucosamine residues in PGN and between N-acetylglucosamine residues in chitin oligomers, thus closely resembling metazoan lysozymes. LYS1-generated fragments trigger immunity-associated responses in a PGN receptor-dependent manner. Activation of defenses has been further shown to occur in the two plants (Arabidopsis and rice) for which PGN perception systems have been described to date (*Willmann et al., 2011*; *Liu et al., 2012a*). Importantly, Arabidopsis plants with strongly reduced *LYS1* expression were impaired in immunity to bacterial infection, suggesting strongly that LYS1 function is an important element of the immune system of this plant. Notably, immunocompromised phenotypes in *LYS1^KD* plants were comparable to those observed in either *lym1 lym3* or *cerk1* PGN receptor mutant genotypes (*Willmann et al., 2011*). We further found that plants overexpressing LYS1 were also susceptible to bacterial infections, suggesting that defined LYS1 levels in wild-type plants are required for LYS1 immune function. The most compelling explanation for this phenotype is that PGN hyperdegradation (in *LYS1^OE* plants) or lack of PGN degradation (in *LYS^KD* mutants) are equally disadvantageous to plant immunity and that immune

*Figure 9. Continued*

The following figure supplements are available for figure 9:

**Figure supplement 1**. Impact of weak *LYS1* overexpression.

activation in Arabidopsis requires oligomeric PGN fragments of a particular minimum degree of polymerization (DP). This view is supported by our findings that prolonged digestion of PGN by LYS1 (*Figure 6D*) or by mutanolysin (*Gust et al., 2007*) abolished the immunogenic activity of PGN. Likewise, immunogenic activities of fungal chitin or oomycete glucans have been reported to require defined minimum ligand sizes with a minimum DP of >5 (*Cheong et al., 1991*; *Zhang et al., 2002*). We therefore propose that *LYS1* overexpression might result in PGN fragments of insufficient size, thereby mimicking the physiological status in *LYS1^KD^* mutants lacking major PGN hydrolytic activities.

Plants produce various carbohydrate-degrading hydrolytic enzyme activities, some of which have been implicated in plant immunity to microbial infection, such as glucanases and chitinases (*van Loon et al., 2006*). While it is often not entirely clear how these enzymes contribute to plant immunity, it is widely assumed that this is due to microcidal activities of these proteins. In our study we have shown that Arabidopsis LYS1 cleaves O-glycosidic bonds formed between GlcNAc (indicative of chitinolytic activity) as well as those formed between GlcNAc and MurNAc (indicative of peptidoglycanolytic activity). However, we have been unable to demonstrate any deleterious effect of LYS1 overexpression on fungal infections, suggesting that *B. cinerea* and *A. brassicicola* at least are not affected by LYS1 function. Likewise, we have been unable to demonstrate direct bactericidal activity of LYS1 to *P. syringae* (not shown), suggesting that the positive role of LYS1 in plant immunity to bacterial infection is not due to its direct inhibitory effect on bacterial fitness. This view is further supported by the fact that *LYS1^OE^* plants with strongly enhanced PGN hydrolytic activity do not exhibit enhanced immunity to *Pseudomonas* infections but become hypersusceptible to infection (*Figure 9*). We cannot rule out at this point LYS1-mediated bacterial lysis, which would likely also result in the release of immunogenic PGN fragments. We would like to emphasize, however, that our findings are in agreement with a predominant role of LYS1 in the generation of PGN fragments that subsequently can trigger plant immunity via PRRs. Hence, plant LYS1 functionally resembles recently described mammalian lysozymes that were shown to generate soluble PGN fragments for PGN receptor NOD2, thereby mediating immunity to *S. pneumoniae* infection in mice (*Davis et al., 2011*).

*LYS1* gene expression is strongly enhanced upon PAMP administration or bacterial infection while expression levels in naive plants are low. It is conceivable that the low constitutive LYS1 levels are sufficient to generate soluble PGN fragments from bulk PGN-containing bacterial walls which are then perceived via the LYM1-LYM3-CERK1 receptor complex. It is possible that the pathogen-inducible later increase in LYS1 activity could have further roles for generating diffusible signals that might serve innate immune activation, not only in cells that are directly in contact with invading microbes but also in cell layers adjacent to infection sites.

A role for plant glycosyl hydrolases in immunogenic PAMP generation and immune activation has been proposed previously (*Mithöfer et al., 2000*; *Fliegmann et al., 2004*). An extracellular soluble bipartite soybean glucan binding protein (GBP) was shown to harbor 1,3-β-glucanase activity and binding activity for glucan fragments of DP >6 derived from intact glucans. Complex glucans constitute major constituents of various *Phytophthora* species, many of which are plant pathogens (*Kroon et al., 2011*). It was therefore suggested that, during infection, GBP endoglucanase activity produces soluble *Phytophthora*-derived oligoglucoside fragments as ligands for the high-affinity binding site within this protein (*Fliegmann et al., 2004*). While this study supported the concept of plant hydrolases tailor-making ligands for plant PRRs, causal evidence for the involvement of the endoglucanase activity in plant immunity was not provided.

Eukaryotic PGN recognition proteins (PGRP, PGLYRP) are conserved from insects to mammals, bind PGN, and function in antibacterial immunity (*Cho et al., 2005*; *Bischoff et al., 2006*; *Dziarski and Gupta, 2010*; *Kurata, 2010*, *2014*). Some PGRP family members are non-enzymatic PRRs (NOD1, NOD2) while others possess PGN-degrading activities (*Gelius et al., 2003*; *Wang et al., 2003*; *Bischoff et al., 2006*; *Dziarski and Gupta, 2010*; *Kurata, 2010*). PGN hydrolytic enzyme activities such as lysozymes have been ascribed functions in direct bacterial killing (*Cho et al., 2005*) and in generating soluble PGN fragments as ligands for PRRs (*Wang et al., 2006*; *Davis et al., 2011*). LYS1 constitutes the first plant lysozyme-type activity for which a role in host immunity has been established.

LYS1 is capable of generating immunogenic fragments from complex PGNs, which themselves serve as ligands for the LYM1-LYM3-CERK1-PGN recognition complex in Arabidopsis. It is noteworthy that LYM1 and LYM3 are PGN recognition proteins that lack apparent intrinsic PGN-degrading activity. We conclude that metazoans and plants employ hydrolytic activities for the decomposition of bacterial PGNs during host immune activation. In addition to the established role of PGNs in pattern-triggered immune activation, host-mediated degradation of bacterial PGNs constitutes another conserved feature of innate immunity in both lineages. However, as the molecular components involved differ structurally among phyla, both facets of PGN-mediated immunity might have evolved convergently.

## Materials and methods

### Plant growth conditions and infections

*A. thaliana* Columbia-0 wild-type and *N. benthamiana* plants were grown on soil as previously described (*Brock et al., 2010*). T-DNA insertion lines for *LYS1* (*lys1-1*, WiscDsLox387C11; *lys1-2*, SALK_095362; *lys1-3*, CSHL_ET14179) were obtained from the Nottingham Arabidopsis Stock Centre. The transgenic *pPR1::GUS* and *secGFP* lines and the *lym1 lym3* and *cerk1-2* mutants have been described previously (*Shapiro and Zhang, 2001*; *Teh and Moore, 2007*; *Willmann et al., 2011*). Rice (*Oryza sativa*) suspension cell cultures were grown in MS-medium (4.41 g/l MS salt, 6% [wt/vol] sucrose, 50 mg/l MES, 2 mg/l 2,4-D) at 150 rpm and sub-cultured every week. Bacterial strains *P. syringae* pv. *tomato* DC3000 or *Pto* DC3000 *ΔAvrPto/AvrPto*, *A. brassicicola* isolate MUCL 20297, and *B. cinerea* isolate BO5-10 were grown and used for infection assays on Arabidopsis leaves of 4–5-week-old plants as described previously (*Lin and Martin, 2005*; *Kemmerling et al., 2007*). To visualize plant cell death and fungal growth on a cellular level, infected plants were stained with Trypan blue in lactophenol and ethanol as described elsewhere (*Kemmerling et al., 2007*).

### Materials

Flg22 peptide has been described previously (*Felix et al., 1999*). The purification of *P. syringae* pv. *tomato* PGN was performed as described previously (*Willmann et al., 2011*). *M. luteus* cell wall preparations and *B. subtilis* PGN were purchased from Invivogen (San Diego, California, United States), Cecolabs (Tübingen, Germany), and Sigma-Aldrich (Hamburg, Germany). PGNs and LPS (from *P. aeruginosa*, Sigma-Aldrich) were dissolved in water at a concentration of 10 mg/ml and stored at −20°C. Mutanolysin was purchased from Sigma-Aldrich.

### Constructs and transgenic lines

Recombinant His6-LYM1 and His6-LYM3 were expressed in *E. coli* and purified as previously described (*Willmann et al., 2011*). As negative control, a protein purification using non-induced cultures harboring the His6-LYM3 construct was performed.

For the *p35S::LYS1* fusion constructs, a 903 bp fragment of the *LYS1* coding sequence without STOP codon was cloned using the primers At5g24090gatF and At5g24090gatR (*Table 1*). In a second PCR, the recombination sites of the inserts were completed using the Gateway adaptor primers attB1 and attB2 (Invitrogen, Darmstadt, Germany). The resulting fragments were then subcloned into pDONR201 (Invitrogen) by using the BP clonase reaction according to the manufacturer's protocol (Invitrogen) and inserted into the binary expression vectors pK7FWG2.0 (C-terminal GFP-tag) (*Karimi et al., 2002, 2005*) or pGWB17 (C-terminal myc-tag) (*Nakagawa et al., 2007*) by using the LR clonase reaction following the manufacturer's protocol (Invitrogen). For the *pLYS1::GUS* reporter construct, a 1948 bp fragment of the *LYS1* promoter sequence was amplified from Arabidopsis Col-0 genomic DNA using the primers At5g24090gatF2 and At5g24090gatR2 (*Table 1*), extended in a second PCR with Gateway adaptor primers attB1 and attB2 and subcloned into pDONR207 (Invitrogen) before being inserted into the binary expression vector pBGWFS7 (*Karimi et al., 2002, 2005*).

For the generation of *pLYS1::GUS* and *p35S::LYS1-GFP* overexpression lines (*LYS1^OE^*), wild-type Col-0 plants were transformed. Stable transgenic lines were generated using standard *Agrobacterium tumefaciens*-mediated gene transfer by the floral dip procedure (*Clough and Bent, 1998*). Expression of GFP fusion proteins was confirmed by immunoblot analysis using an anti-GFP antibody (Acris Antibodies GmbH) and anti-tobacco class III chitinase antibody (kindly provided by Michel Legrand, IBMP Strasbourg, France). The histochemical detection of β-glucuronidase (GUS) enzyme activity in whole leaves of *pLYS1::GUS* or *pPR-1::GUS* transgenic Arabidopsis (*Shapiro and Zhang, 2001*) was determined as described earlier (*Gust et al., 2007*).

**Table 1.** Primers used in this study

| AGI | Primer name | Sequence 5′ → 3′ |
|---|---|---|
| At5g24090 (LYS1) | At5g24090F1 | CCAGAGGTGGCATAGCCATC |
| | At5g24090R1 | CATCTGGTGGGATATAGCCAC |
| | At5g24090F | ATGACCAACATGACTCTTCG |
| | At5g24090R | TCACACACTAGCCAATATAG |
| | At5g24090RP2 | TGATGCCACGAGACTGAC |
| | LP_N853931 | TGACGAACCATGATAAATGGG |
| | RP_N853931 | CATAACCTCACACTGTGCTCG |
| | LP_N595362 | TAGTGCATGCATGTTAAACCG |
| | RP_N595362 | AGCTCCTCAATGTCCATTTCC |
| | Salk-Lba | TGGTTCACGTAGTGGGCCATCG |
| | Ds5-1 | GAAACGGTCGGGAAACTAGCTCTAC |
| | Wisc-Lba (p745) | AACGTCCGCAATGTGTTATTAAGTTGTC |
| | At5g24090Fq | CACTTGCACCCATTTTGGC |
| | At5g24090Rq | CCTCGACCCAATCGAGTA |
| | At5g24090miR-s | GATTTGACGTAAGCATACCGCCCTCTCTCTTTTGTATTCC |
| | At5g24090miR-a | GAGGGCGGTATGCTTACGTCAAATCAAAGAGAATCAATGA |
| | At5g24090miR*s | GAGGACGGTATGCTTTCGTCAATTCACAGGTCGTGATATG |
| | At5g24090miR*s | GAATTGACGAAAGCATACCGTCCTCTACATATATATTCCT |
| | At5g24090gatF | AAAAAGCAGGCTACATGACCAACATGACTCTTCG |
| | At5g24090gatR | AGAAAGCTGGGTACACACTAGCCAATATAGATG |
| | At5g24090gatR-STOP | AGAAAGCTGGGTATCACACACTAGCCAATATAG |
| | At5g24090gatF2 | AAAAAGCAGGCTATGCCGTAGGCGAGTGTTTC |
| | At5g24090gatR2 | AGAAAGCTGGGTGTTTTTGGTTAAAGATGTTTG |
| At1g07920/30/40(EF1α) | Ef1α-100-f | GAGGCAGACTGTTGCAGTCG |
| | Ef1α-100-r | TCACTTCGCACCCTTCTTGA |
| At2g19190 (FRK1) | FRK1-F | AAGAGTTTCGAGCAGAGGTTGAC |
| | FRK1-R | CCAACAAGAGAAGTCAGGTTCGTG |
| At4g02540 | At4g02540-qf1 | GTACCACGCCTATCTATT |
| | At4g02540-qr1 | CTCATAGAAGAAACCAGCA |
| At1g05615 | At1g05615-qf1 | GGATTCCTATCTCTACCT |
| | At1g05615-qr1 | TTCTTTACCCTCATCAACC |
| At5g58780 | At5g58780-qf1 | CTCTCTTCTCTTTTATCTCTCC |
| | At5g58780-qr1 | CTCCTCCACTCCTACCACA |
| At3g51010 | At3g51010-qf1 | GCGTCGTGCTTTTATACTG |
| | At3g51010-qr1 | TTCTTCCTCTTCGCCTCT |
| At1g21880 (LYM1) | Lym1-100-f | TACAACGGTATAGCCAACGGCACT |
| | Lym1-100-r | GTGGAGCTAGAAGCGGCGCA |
| At1g77630 (LYM3) | Lym3-100-f | ACTTCGCAGCAGAGTAGCTC |
| | Lym3-100-r | AGCGGTGCTAATTGTTGCGG |
| At3g21630 (CERK1) | CERK1-100-f | GGGCAAGGTGGTTTTGGGGCT |
| | CERK1-100-r | CCGCCAAGAACTGTTTCGATGCC |
| | attB1 | GGGGACAACTTTGTACAAAAAAGCAGGCT |
| | attB2 | GGGGACCACTTTGTAC AAGAAAGCTGGGT |

Artificial microRNA-mediated gene silencing was used to specifically knock down *LYS1* in the Col-0 background as mutant lines carrying T-DNA insertions in the *LYS1* gene were unavailable. The Web microRNA Designer (WMD; http://wmd.weigelworld.org) was used to select the primers At5g24090miR-s, At5g24090miR-a, At5g24090miR*s, and At5g24090miR*s (*Table 1*) for the generation of an artificial 21mer microRNA (*Schwab et al., 2005*). The *LYS1*-specific amiRNA was then introduced into the vector miR319a pBSK (pRS300) by directed mutagenesis. Knock-down of the *LYS1* transcript level in stably transformed Col-0 plants (*LYS1* knock-down line, *LYS1^KD^*) was determined by RT-qPCR using primers At5g24090Fq and At5g24090Rq listed in *Table 1*. Off-target genes were identified using the Web microRNA Designer and transcript levels of the four top hits were determined by RT-qPCR using primers listed in *Table 1*.

## Transient protein expression

*A. tumefaciens*-mediated transient transformation of *N. benthamiana* was performed as described previously (*Brock et al., 2010*). The leaves were examined for expression of tagged fusion proteins 3–4 days post infection. Expression of fusion proteins was confirmed by immunoblot analysis using anti-myc antibodies (Sigma-Aldrich) and localization studies of GFP fusion proteins were carried out using a confocal laser-scanning microscope, as described elsewhere (*Willmann et al., 2011*).

## LYS1 purification from *LYS1^OE^* plants

From 5-week-old *LYS1^OE^* Arabidopsis plants, 100 g leaf tissue was frozen in liquid nitrogen and ground to fine powder. After addition of buffer A (20 mM sodium acetate, pH 5.2, 0.01% [vol/vol] β-mercaptoethanol), the extract was incubated on ice overnight. After filtration through four layers of cheesecloth, the homogenate was centrifuged at 10,000× g for 30 min. The supernatant was loaded on a cation exchange column (SP Sepharose, GE Healthcare, München, Germany) equilibrated with buffer A. The column was washed with buffer A and proteins were eluted with a 0 to 1 M NaCl gradient in buffer A. The elution fractions were monitored for LYS1 activity with the 4-MUCT assay and protein purification was further confirmed by SDS-PAGE. 4-MUCT-active fractions were pooled and exchanged to buffer A using Vivaspin 3 kDa columns (GE Healthcare). Protein concentration was determined using the Bradford assay.

For LC-MS analysis, the Coomassie Blue-stained band of the major cleavage product of the purified LYS1-GFP sample was cut and in-gel digested with trypsin, as described elsewhere (*Borchert et al., 2010*). LC-MS analyses of the peptides were done on an EasyLC nano-HPLC (Proxeon Biosystems) coupled to an LTQ Orbitrap Elite mass spectrometer (Thermo Scientific) as described elsewhere (*Conzelmann et al., 2013*). MS data were processed using the software suite MaxQuant, version 1.2.2.9 (*Cox and Mann, 2008*) and searched using Andromeda search engine (*Cox et al., 2011*) against a target-decoy *A. thaliana* database containing 33,351 forward protein sequences, the sequence of the LYS1-GFP fusion protein, and 248 frequently observed protein contaminants. MS data were processed twice, once considering only fully tryptic peptides and once considering only semi-tryptic peptides. In each case, two missed cleavage sites were allowed, carbamidomethylation of cysteine was set as the fixed modification, and N-terminal acetylation and methionine oxidation were set as variable modifications. Mass tolerance was set to 6 parts per million (ppm) at the precursor ion and 20 ppm at the fragment ion level. Identified peptide spectrum matches (PSM) were statistically scored by MaxQuant software by calculation of posterior error probabilities (PEP) (*Käll et al., 2008*) for each PSM. All PSMs having a PEP below 0.01 were considered as valid.

For matrix-assisted laser desorption/ionization time-of-flight mass spectrometry (MALDI-TOF-MS), protein digestion was performed as described elsewhere (*Maurer et al., 2013*; *Amin et al., 2014*). Briefly, the Coomassie Blue-stained band of the major cleavage product of the FPLC-purified LYS1-GFP sample was cut from the gel and destained with 30% (vol/vol) acetonitrile in 50 mM ammonium bicarbonate buffer. Disulfide bonds were reduced with 10 mM dithiothreitol (DTT), 50 mM iodoacetamide was used to alkylate the cysteines followed by overnight protein digestion with mass spectrometry grade trypsin (Promega, Manheim, Germany) at 37°C. The digests were acidified by the addition of trifluoric acid (TFA) to a final concentration of 0.5%. Extracted peptides were desalted and mixed with an equal volume of 2,5-dihydroxybenzoic acid for Reflex-IV MALDI-TOF-MS (Bruker Daltonics, Bremen, Germany) measurements. Each spectrum was processed internally for trypsin autolysis before database search. The identity of protein was annotated using the SwissProt database (542782 sequences; 193019802 residues). To achieve the best possible results, the search parameters were as follows: one miscleavage was set for trypsin specificity and carbamidomethyl modification of cysteine and oxidation of methionine were selected as fixed and optional modifications, respectively. At

a mass tolerance of 5 ppm, only protein scores greater than 70 (p<0.05) were assigned significant with an expected value of $10^{-7}$.

## Protein extraction and enzymatic assays

Apoplastic washes were obtained from mature leaves of 4-week-old Arabidopsis plants by vacuum-infiltrating complete rosettes with 20 mM sodium acetate, pH 5.2. Afterwards, leaf tissue was dipped dry on paper towels, placed in 50 ml Falcon tubes and spun at 1000× g for 5 min at 4°C. Collected fluids were concentrated tenfold using Vivaspin 500 columns with a 3 kDa cut-off (GE Healthcare).

Isolation of mesophyll protoplasts from leaves of 4–5-week-old Arabidopsis plants was performed according to a protocol described previously (*Yoo et al., 2007*). Isolated protoplasts were resuspended in W5 solution (2 mM MES, pH 5.7, 154 mM sodium chloride, 125 mM calcium chloride, 5 mM potassium chloride) and incubated overnight at room temperature in the dark ($2 \times 10^5$ protoplasts in 1 ml W5 solution). Subsequently, protoplasts were removed by centrifugation (20 s, 800 rpm, 4°C) and secreted proteins in the medium were concentrated using Vivaspin 2 columns with a 10 kDa cut-off (GE Healthcare).

Total protein extracts from the harvested protoplast pellet of 4–5-week-old leaves of *A. thaliana* or *N. benthamiana* were prepared using 20 mM sodium acetate, pH 5.2, supplemented with 15 mM β-mercaptoethanol and proteinase inhibitor cocktail (Roche Applied Science, Mannheim, Germany). Approximately 40–60 µg total protein of the leaf extracts or 15 µg of the protoplast samples were added to the enzyme assays. For all in-tube enzyme assays described in the supplemental information, the reaction mix was incubated with shaking at 37°C in 20 mM sodium acetate, pH 5.2.

The 4-MUCT chitinase assay was performed as described (*Brunner et al., 1998*). Briefly, the hydrolytic activity towards 4-MUCT (Sigma-Aldrich) was measured for 30 min and compared with that of 2 µg *S. griseus* chitinase (Sigma-Aldrich). After enzyme incubation in 250 µl final volume of 0.05% (wt/vol) 4-MUCT, 20 µl of the reaction mixture were removed and added to 980 µl 0.2 M sodium carbonate solution. Free 4-MU (Sigma-Aldrich) was used for the generation of a standard curve. The intensity of the fluorescence was monitored with an MWG Sirius HT fluorescence microplate reader. For the zymogram, discontinuous cetyltrimethylammonium bromide (CTAB) polyacrylamide gel electrophoresis was performed using a 12% separating gel (43 mM potassium hydroxide [KOH], 280 mM acetic acid, pH 4.0, 12% [vol/vol] acrylamide bisacrylamide 37.5:1, 8% [vol/vol] glycerol, 1.3% ammonium persulphate and 0.16% N, N, N, N-tetramethylethylene diamine [TEMED]) overlaid by a 4% stacking gel (64 mM KOH, 94 mM acetic acid, pH 5.1, 4% acrylamide, 1.25% ammonium persulphate and 0.125% TEMED). Prior to loading, the gel was pre-run using anode buffer (40 mM beta-alanine, 70 mM acetic acid, 0.1% CTAB, pH 4.0) and cathode buffer (50 mM KOH, 56 mM acetic acid, pH 5.7, 0.1% CTAB) for 1 hr at 250 Volts. Crude protein extracts were mixed with an equal volume of loading buffer (5 M urea, 25 mM potassium acetate, pH 6.8, methylene blue) and separated for 2 hr at 150 Volts and 4°C. After electrophoresis the CTAB gel was washed with 20 mM sodium acetate, then sprayed with 0.00625% (wt/vol) 4-MUCT in 20 mM sodium acetate, pH 5.2, and incubated at 37°C for 30 min. Fluorescent bands were documented under UV light using the Infinity-3026WL/26MX gel imaging system (PeqLab, Erlangen, Germany).

The turbidity assay was done as described previously (*Park et al., 2002*). Lytic activity towards *M. luteus* cell wall preparations or *B. subtilis* peptidoglycan (Invivogen, Cecolabs) was measured for 4 hr and compared with that of 1 µg hen egg-white lysozyme (Sigma-Aldrich). 1 ml 0.02% (wt/vol) *M. luteus* cells or PGN suspension were incubated together with the enzyme and the decrease in absorbance at 570 nm of the suspension was measured with a spectrophotometer over time.

The 4-MUC cellulase assay was performed using 4-methylumbelliferyl-β-D-cellobioside (4-MUC; Sigma-Aldrich) as substrate. 1 mM 4-MUC was incubated in 20 mM sodium acetate (pH 5.2) at 37°C for 1 hr in a 96 well plate with either 40 µg purified LYS1 or cellulase (Duchefa, Haarlem, The Netherlands) in a total volume of 100 µl. The reaction was stopped with 0.2 M sodium carbonate and the intensity of the fluorescence was monitored with an MWG Sirius HT fluorescence microplate reader using excitation and emission wavelengths of 365 nm and 455 nm, respectively.

## HPLC analysis

500 µg/ml *B. subtilis* PGN was incubated with 140 µg LYS1 purified from *LYS1^{OE}* plants or controls in 20 mM sodium acetate, pH 5.2, at 37°C with shaking for 7 hr. After stopping the reaction by heating at 100°C for 10 min, the reaction was centrifuged and the supernatant analyzed by HPLC. The analyses were done by Cecolabs on an Agilent 1200 system with a Prontosil C18-RP column (Bischoff Chromatography, Leonberg, Germany). The mobile phase was (A) 100 mM sodium phosphate, 5% (vol/vol) methanol and (B) 100 mM sodium phosphate, 30% (vol/vol) methanol.

## Immune responses

RNA isolation, semi-quantitative RT-PCR and RT-qPCR analysis were performed as described previously (*Kemmerling et al., 2007*; *Willmann et al., 2011*). For RT-qPCR, all quantifications were made in duplicate on RNA samples obtained from three independent experiments, each performed with a pool of 3–5 seedlings or two leaves. *EF1α* transcripts served normalization; corresponding water controls were set to 1. The sequences of the primers used for PCR amplifications are given in *Table 1*. The histochemical detection of β-glucuronidase (GUS) enzyme activity in whole leaves of *pLYS1::GUS* or *pPR-1::GUS* transgenic Arabidopsis (*Shapiro and Zhang, 2001*) was determined as described earlier (*Gust et al., 2007*). For the measurement of extracellular pH, 300 µl of cultured rice cells were transferred to 48 well plates and equilibrated at 150 rpm for 30 min. After addition of elicitors, the pH in the cell culture was monitored with an InLab Micro electrode (Mettler Toledo, Gießen, Germany).

For assays with LYS1-digested PGN, 100 µg/ml *B. subtilis* PGN was incubated with 40 µg LYS1 purified from *LYS1^{OE}* plants or controls in 2.5 mM MES, pH 5.2, at 37°C with shaking for 4 hr. After stopping the reaction by heating at 100°C for 10 min, the reaction was centrifuged and the supernatant used for triggering immune responses.

## Statistical methods

Statistical significance between two groups has been checked using the Student's *t* test. Asterisks represent significant differences (*$p<0.05$; **$p<0.01$; ***$p<0.001$). One-way analysis of variance (ANOVA) was performed for multiple comparisons combined with Duncan's multiple range test indicating significant differences with different letters ($p<0.05$).

## Acknowledgements

We thank Andreas Kulik and Friedrich Götz for bacterial fermentation and Gary Stacey and Michel Legrand for providing the anti-CERK1 and anti-class III chitinase antibody, respectively.

## Additional information

### Competing interests

TN: Reviewing editor, *eLife*. The other authors declare that no competing interests exist.

### Funding

| Funder | Grant reference number | Author |
| --- | --- | --- |
| Deutsche Forschungsgemeinschaft | SFB 766 | Xiaokun Liu, Heini M Grabherr, Roland Willmann, Dagmar Kolb, Ute Bertsche, Thorsten Nürnberger, Andrea A Gust |

The funders had no role in study design, data collection and interpretation, or the decision to submit the work for publication.

### Author contributions

XL, HMG, Conception and design, Acquisition of data, Analysis and interpretation of data; RW, DK, UB, DK, MF-W, BA, Acquisition of data, Analysis and interpretation of data; FB, Conception and design, Drafting or revising the article; GF, MO, Drafting or revising the article, Contributed unpublished essential data or reagents; TN, AAG, Conception and design, Analysis and interpretation of data, Drafting or revising the article

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
