## [Decision Letter]

Thank you for sending your work entitled “Bacterial cell wall decomposition mediates pattern-triggered immunity in Arabidopsis” for consideration at *eLife*. Your article has been favorably evaluated by a Senior editor and 3 reviewers, one of whom is a member of our Board of Reviewing Editors.

The following individuals responsible for the peer review of your submission have agreed to reveal their identity: Jean Greenberg (Reviewing editor).

The Reviewing editor and the other reviewers discussed their comments before we reached this decision, and the Reviewing editor has assembled the following comments to help you prepare a revised submission.

The authors are interested in whether plants help generate soluble MAMPs from pathogens that can then be detected by plant receptors to induced defenses. They focus on LYS1, which they show can generate PGN fragments; the LYS1 gene is induced by various MAMPs and “Non-adapted” or non-pathogenic *P. syringae*. Recombinant LYS1 is not active, so they overexpress LYS1-GFP in plants and use this as a source for enzyme to characterize. They also characterize the overexpression plants and knockdowns. The biochemical effects are for the most part convincing but might be more so if they were presented as specific activities with some kinetic analyses that were compared with the known enzymes they use. A major finding is that the plants with OE and KD seem to have the same phenotypes with respect to hypersusceptibility to *P. syringae* and lack of MAMP response (at least FRK1 expression) with PGN treatment. Several very important issues came up in the review process that need to be addressed:

1) Claims of specificity: Since LYS1 has chitinolytic activity, there is the potential that this activity could be playing a role in anti-fungal defense. Although the authors tested this, they can’t exclude that with certain fungal isolate/host combinations LYS1 might have a role in fungal defense. Therefore, we advise toning down the claim of specificity for responses to bacteria.

2) Organization and analysis methods: Figure panels should be organized in the order they are discussed. Add something about how many times each experiment was replicated. Statistical analysis needs to be improved. In complex experiments where multiple comparisons can/should be made, use ANOVA and multiple comparison post hoc test. It’s especially difficult to understand the comparisons made in 3E and Figure 5.These data should all be analysed together and grouped into significance groups.

3) Concerning the KD and overexpression plants: the specificity of the artificial miRNA needs to be established. The authors need to show that only the LYS1 gene is being silenced and not other genes that may have some sequence similarity. Because the results of Figure 8 are confusing, it would be good to use a knockout plant such as a SALK mutant (e.g. SALK_095362). Figure 2 is the only example where the KD seems to affect the relative PGN hydrolysis activity. Therefore, it’s especially important to do the correct statistical analysis. The effect of the KD would be more convincing if we had conditions where the level of protein in WT can be detected (or maybe one of the bands in D in the Col lane is right?). Did the authors try using apoplast washes to enrich for the activity? How about inducing conditions (e.g. MAMP treatment?) Since LYS expression is normally PAMP-induced, it is important to show that a PAMP-induced increase in PGN-degrading activity is decreased in extracts from *LYS1-KD* lines, when compared to extracts from WT plants.

The overexpression lines with 300 fold higher than wild type seems way too high (Figure 2) and could be causing unforeseen effects. At a minimum the authors should check that the consequence of OE is not causing downregulation of the receptors that perceive the PGNs. If this were the case, it would change the interpretation of the experiments. Some lines where the overexpression is more modest would be helpful for determining if increased resistance can be achieved. In any case with Figure 8, the authors should check the kinetics to see if there is a time point at which the response of OE is faster/higher than WT.

4) Localization: The experiments performed in protoplasts (Figure 3) is not sufficient to claim that LYS1 localizes to the apoplast, as it is highly likely that the protoplast medium also contains damaged protoplasts. The authors should transiently express LYS1-GFP in *N. benthamiana*, and show by immunoblot analysis that LYS1-GFP is enriched in apoplastic fluids after PAMP treatment. This experiment should be very straight-forward in this system.

5) Consequence of down and up-regulation for susceptibility to infection: The results of Figure 8 are very puzzling. They show that over-expression lines have the same phenotype, with regard to bacterial virulence, as the silenced lines. This is not due to co-suppression since the authors measured significant levels of both mRNA and protein levels in these over-expression lines. In the Discussion, the authors postulate that higher levels of LYS1 results in a more extensive hydrolysis of the PGN resulting in the loss of the correct sized fragment for induction of innate immunity. This seems a reasonable hypothesis but one that requires experimental validation. This requires the use of a quantitative assay for the immune response that can be followed over time. There are many such relatively easy assays that could be used, such as ROS production or gene expression. The authors should test whether the *LYS1* OE and KD lines differ in their temporal response to both PGN and bacterial inoculation using such assays. If their hypothesis is correct, then one might predict that the OE lines would show some response very early but this would decline rapidly as the elicitor is completely hydrolyzed. One might also predict, for example, that adding higher levels of PGN elicitor might elicit a response on the OE lines (i.e., one might be able to saturate the enzyme and slow hydrolysis to the point where elicitation can be measured).

6) Association of PGN-degrading activity with LYS1. The authors must demonstrate that the PGN-degrading activity detectable upon over-expression of *LYS1* is indeed due to LYS1 itself, and not to another enzyme whose expression or activity is increased by *LYS1* over-expression. For this, the authors should transiently express a catalytically-inactive variant of LYS1 (based on the knowledge on similar enzymes in other kingdoms) and compare the PGN-degrading activity of the corresponding extract when compared to an extract from leaves over-expressing WT LYS1.

7) What is LYS1 generating? In Figure 5, the authors say that the mutants don't respond to the fragment generated, but the data shows there is some response. In fact, it looks like the basal FRK1 level with buffer is much lower in the mutants than in the Col-O. Therefore, are the fold inductions are retained and just the basal responses are altered? As mentioned above, all the data should be analysed together for significance groups. LYM1, LYM3 and CERK1 genes are essential for PGN signal transduction, so the mutants should have almost no response to soluble PGN. So it’s not clear whether this LYS1-generated peptidoglycan (PGN) fragments is soluble PGN or not, if *cerk1* mutants are still responding to it.

Other issues to address:

Figure 1 text says suspensions of Micrococcus luteus cell but legend and methods say it’s really cell wall prep/extract. It’s not clear what the negative control is. This should be documented in the Methods.

Figure 1 100 µg PGN but 100 µg/ml LPS. Need consistent description (per ml )

Figure 2: how can the authors be sure that the fast-migrating band corresponds to the full-length LYS1 protein, or actually, event to LYS1. Did the authors attempt to confirm the identity of this band by mass spectrometry?

In Figure 3, the relative hydrolytic activity of LYS1 was calculated using chitinase as a standard. It would be better if the authors could directly measure LYS1 kinetic parameters (e.g., Kd). This would be an important measurement since it would get directly to the issue of physiological relevance. These are standard techniques, which should be within the authors' ability given their access to purified LYS1 protein.

In Figure 2, there is another band which located between 35Kd and 40 Kd shown in LYS1OE samples, please clarify what it is. If it is possible, please do this experiment again; the picture looks very bad. What are the possible effects of proteolysis on the protein, which seems essential for its purification? To be thorough, the authors should identify the cleavage site so that they have a better idea of how much of the LYS1 protein remains. For example, could proteolysis be eliminating a key regulatory portion of the protein?

[Editors' note: further clarifications were requested prior to acceptance, as described below.]

The authors did a good job trying to respond to the comments to improve the manuscript. They added analysis of an additional O/E line that does not seem to cause statistically significant increase in *P. syringae* growth after 2 days; this line shows only moderate overexpression of *LYS1*. However, there are a few issues with part of the analysis that need to be addressed before final acceptance.

1) The authors need to harmonize the description of the statistical analysis with what is presented in the figures. The authors say they added multiple comparison tests, but the way they describe these is confusing. For example in Figure 4, if the multiple comparison test was done, then all values in the panel should be labeled by grouping into like categories (i.e. A, B etc). Similarly in 6B (or 6C), significance at P<0.001 are given, but we cant tell whether LYS1 is different from mutanolysin or whether all the treatments in the mutants are giving the same or different values. If all the data was analyzed together, you should be able to label each group (all values that are similar get a different letter).

2) For the P. syringae infection experiments, the authors say in the discussion that overexpression or down-regulation of LYS1 causes the same magnitude of defect. Actually, they don't do any multiple comparison test of the data in Figure 9 and it looks like the O/E are even more susceptible than the down-regulated plants. The authors should use multiple comparison test here to show whether the effects are the same or not. A couple of other things about the description of this experiment. (1) the title of the Figure 9 should be changed, since there is no “mutation of LYS1”. (2) This experiment does not measure the growth rate since only one endpoint is taken. Therefore no conclusion about the rate can be given.

3) The authors tested the transcript level of receptors that would detect PGN fragments liberated by LYS1 in the *LYS1* overexpression plants; this is important for excluding the possibility that high overexpression of *LYS1* has unintended consequences on the receptor levels. A more informative experiment would be to check protein levels, as this is the most relevant assay (transcript levels don't always reflect protein levels). There is an available antibody for CERK1- did the authors try using this antibody?

---

## [Author Response]

*1) Claims of specificity: Since Lys1 has chitinolytic activity, there is the potential that this activity could be playing a role in anti-fungal defense. Although the authors tested this, they can’t exclude that with certain fungal isolate/host combinations LYS1 might have a role in fungal defense. Therefore, we advise toning down the claim of specificity for responses to bacteria*.

As suggested by the Reviewers we toned down the claim of specificity for responses to bacteria.

*2) Organization and analysis methods: Figure panels should be organized in the order they are discussed. Add something about how many times each experiment was replicated. Statistical analysis needs to be improved. In complex experiments where multiple comparisons can/should be made, use ANOVA and multiple comparison post hoc test. It’s especially difficult to understand the comparisons made in 3E and*
Figure 5*.These data should all be analysed together and grouped into significance groups*.

All figure panels, particularly in Figure 2, are now discussed in the text as they are appearing in the figure(s). Each figure legend now also includes a statement about how many repetitions were performed for the experiments.

As suggested by the Referees, we now analyzed complex data shown in Figures 4 and 6 (corresponds to old Figures 3 and 5) using one way analysis of variance (ANOVA) followed by Turkey’s multiple comparison post hoc test. This is now stated in the respective figure legends and in the Methods section. Notably, the ANOVA analysis validated significant differences obtained initially with the Student’s t-test with the only exception in Figure 4, now showing no significant differences between supernatant samples from wild type and *LYS1*^*KD*^ lines (also see Reply 3.3.). Thus, in this case we deleted the corresponding statement from the main text.

*3.1) Concerning the KD and overexpression plants: the specificity of the artificial miRNA needs to be established. The authors need to show that only the LYS1 gene is being silenced and not other genes that may have some sequence similarity*.

To test specificity of the target sequence used for the artificial miRNA construct, we performed qPCR analyses of potential off-target genes. BLAST searches indicated that no other chitinase gene was a potential off-target. As stated in the Methods section, off-target genes were identified using the Web microRNA Designer (new Figure 3) and transcript levels of the four top hits were determined by qRT-PCR (new Figure 3). In contrast to *LYS1* transcript levels in the *LYS1*^*KD*^ lines, the transcription of potential off-target genes was not affected.

*3.2) Because the results of*
Figure 8
*are confusing, it would be good to use a knockout plant such as a SALK mutant (e.g. SALK_095362)*.

As suggested by the Referees, we initially obtained three independent T-DNA insertion lines, SALK_095362 (*lys1-2*) as suggested by the Referees, and additionally WiscDsLox387C11 (*lys1-1*) and CSHL_ET14179 (*lys1-3*). For all lines we could identify homozygous progeny, however, *LYS1* transcript levels appeared to be like wild type. These results are now presented in Figure 3—figure supplement 1. Consequently, we generated amiRNA lines and only used these KD-lines in our experiments.

*3.3)*
Figure 2
*is the only example where the KD seems to affect the relative PGN hydrolysis activity. Therefore, it’s especially important to do the correct statistical analysis*.

As requested by the Referees, we now applied ANOVA and multiple comparison post hoc test to our data obtained in the protoplast system (now Figure 4). However, differences between wild type supernatant samples and *LYS1*^*KD*^ supernatant samples did not proof to be significantly different in the one-way ANOVA analysis followed by a Turkey multiple comparison post hoc test. Thus, as also stated in Reply 2 we deleted this statement from the main text.

*3.4) The effect of the KD would be more convincing if we had conditions where the level of protein in WT can be detected (or maybe one of the bands in D in the Col lane is right?). Did the authors try using apoplast washes to enrich for the activity? How about inducing conditions (e.g. MAMP treatment?) Since LYS expression is normally PAMP-induced, it is important to show that a PAMP-induced increase in PGN-degrading activity is decreased in extracts from* LYS1-KD *lines, when compared to extracts from WT plants*.

Concerning the detection of native LYS1 in the wild type plants we repeated the western blot formerly shown in Figure 2 (now new Figure 2) but now additionally included protein extracts from one *LYS1*^*KD*^-line. However, we could not detect any differences in the background band pattern between wild type and the *LYS1*^*KD*^-line. Also, in apoplastic fluids we could not detect any protein band in the wild type samples that was absent in the *LYS1*^*KD*^-line (data not shown). Thus wild type LYS1 protein levels are most likely too low to be detected.

As suggested by the Referees, we infected wild type, *LYS1*^*OE*^ and *LYS1*^*KD*^ lines with *Pseudomonas syringae* pv *phaseolicola* (which induces *LYS1* gene expression, Figure 1) and used protein extracts from infected versus uninfected leaves for enzyme assays. However, slight but not significant increases in hydrolytic activities against 4-MUCT could be observed in both the wild type and the *LYS1*^*KD*^ samples and even in the *LYS1*^*OE*^ line. As chitinases belong to the family of pathogenesis-related proteins which are generally induced upon pathogen attack it can be assumed that also other chitinases/hydrolytic activities are induced upon *Pph* infection, overlaying the effect of *LYS1* downregulation.

*3.5) The overexpression lines with 300 fold higher than wild type seems way too high (*Figure 2*) and could be causing unforeseen effects. At a minimum the authors should check that the consequence of OE is not causing downregulation of the receptors that perceive the PGNs. If this were the case, it would change the interpretation of the experiments*.

We acknowledge that *LYS1* overexpression might indeed have side effects. To exclude a direct effect of LYS1-overexpression on PGN receptor expression, we examined transcript levels of *LYM1*, *LYM3* and *CERK1*. However, transcription of these receptor genes was not affected in the *LYS1*^*OE*^ lines and the results are now presented in Figure 9—figure supplement 1.

*3.6) Some lines where the overexpression is more modest would be helpful for determining if increased resistance can be achieved*.

We now include a new line (*LYS1*^*OE*^*-3*) in the *Pseudomonas* infection assays (Figure 9—figure supplement 1). In accordance with ten-fold less increased *LYS1* transcript levels in mature leaves of the *LYS1*^*OE*^-3 line compared to the *LYS1*^*OE*^-1 line, also leaf protein levels of LYS1-GFP and free LYS1 are less abundant in the *LYS*^*OE*^-3 line. Moreover, susceptibility to *Pseudomonas* infection was only slightly but not significantly (P = 0,064, Student’s t-test) increased. It should, however, be noted that differences in bacterial growth smaller than half log are difficult to statistically validate. These results however confirm that lowering *LYS1* expression levels, accompanied by lower LYS1 hydrolytic activity on PGN, brings down these lines close to wild-type.

*3.7) In any case with*
Figure 8*, the authors should check the kinetics to see if there is a time point at which the response of OE is faster/higher than WT*.

As suggested by the Referees we compared the kinetics of early defense responses in wild type, *LYS1*^*OE*^ and *LYS1*^*KD*^ lines. We conducted ion leakage experiments after *Pto* DC3000 infection and determined *FRK1* gene expression after treatment with either 100 or 500 µg/ml PGN (see Figure 10 below). For ion leakage we measured every hour starting 1 hour post infection up to 6 hours and then every 12 hours up to 36 hours, but in three independent experiments we could not observe any differences between the responses in the different lines at these early time points. For the determination of *FRK1* transcript levels we took samples after 30 min, 1 hour, 3 hours and 6 hours. Here, at early time points such as 30 min to 1 hour, *FRK1* gene expression is not really induced by PGN, thus error bars are very big. At time points 3 and 6 hours, however, we observed *FRK1* induction in WT samples but not in *LYS1*^*KD*^ lines, confirming our results shown in Figure 9. Likewise, *LYS1*^*OE*^ samples treated for 6 hours with 100 µg/ml PGN mimicked the lack of *FRK1* gene expression also observed in the *LYS1*^*KD*^ mutant (see also Figure 9). Notably, treatment of *LYS1*^*OE*^ seedlings with 500 µg/ml PGN resulted in *FRK1* gene induction, however, at no time point did we observe a stronger response in *LYS1*^*OE*^ lines than in the wild type. As information for the Referees, these data are included below:Author response image 1.Author response image 1: Early defense responses are not enhanced in *LYS1*^*OE*^-lines.(A) Ion leakage measured in leaves of WT, *LYS1*^*OE*^ or *LYS1*^*KD*^ lines at indicated time points after infiltration of *Pto* DC3000. (B) Determination of *FRK1* transcript levels in seedlings of wild type plants or transgenic *LYS1* plants treated for 30 min, or 1, 3, or 6 hours with 100 or 500 µg/ml *B. subtilis* PGN. Total RNA was subjected to RT-qPCR using *FRK1* specific primers, *EF1α* transcript was used for normalization. Data represent means ± S.D. of triplicate samples.

We would like to emphasize that in the *LYS1*^*OE*^ lines, an enhanced level of LYS1 enzyme activity is present all the time, thus also right at the beginning of both the *Pto* infection and the PGN treatment. Hence, we consider it being very challenging to find a time point at which the *LYS1*^*OE*^ line might show increased immune responses compared to wild type plants.

Moreover, we would like to draw the Reviewers’ attention again to Figure 6 (old Figure 5). Here, both the PGN preparation treated with buffer only AND the PGN preparation treated overnight with LYS1 lack immunogenic activity in the respective supernatant. We conclude from this experiment that a lack of PGN hydrolysis in buffer controls in Figure 6 (old Figure 5) accompanied with no induction of immune responses is reminiscent of the situation in *LYS1*^*KD*^ lines, whereas a complete PGN digestion overnight (new Figure 6), again accompanied with no induction of immune responses, is reminiscent of the situation in *LYS1*^*OE*^ lines.

*4) Localization: The experiments performed in protoplasts (*Figure 3*) is not sufficient to claim that LYS1 localizes to the apoplast, as it is highly likely that the protoplast medium also contains damaged protoplasts. The authors should transiently express LYS1-GFP in* N. benthamiana*, and show by immunoblot analysis that LYS1-GFP is enriched in apoplastic fluids after PAMP treatment. This experiment should be very straight-forward in this system*.

As suggested, we transiently expressed a *p35S::LYS1-GFP* construct in *Nicotiana benthamiana* which resulted in labelling of the cell periphery, whereas expression of a construct lacking the *LYS1* signal peptide-encoding sequence yielded labelling of intracellular structures (Figure 4—figure supplement 1). Use of the fluorescent dye FM4-64, a plasma membrane and early endosome marker (7), revealed that LYS1 signals co-localized to a large extent with the plasma membrane (Figure 4—figure supplement 1). Thus, LYS1 likely operates in close vicinity of the plant surface.

To confirm a LYS1-localization in the apoplast, we prepared apoplastic washes from *LYS1*^*OE*^ Arabidopsis lines. Both the LYS1-GFP fusion protein as well as free LYS1 was detectable in concentrated apoplastic fluids whereas the cytoplasmic protein MPK3 was only present in the total leaf protein samples (Figure 4—figure supplement 1). As expression of the constructs we used was driven by the 35S-promoter we did not see any further enrichment after PAMP treatment (data not shown).

Together with the previous identification of LYS1 within the *Arabidopsis* cell wall proteome ([48], as already stated in the previous manuscript) our data suggest that LYS1 acts in the plant apoplast.

*5) Consequence of down and up-regulation for susceptibility to infection: The results of*
Figure 8
*are very puzzling. They show that over-expression lines have the same phenotype, with regard to bacterial virulence, as the silenced lines. This is not due to co-suppression since the authors measured significant levels of both mRNA and protein levels in these over-expression lines. In the Discussion, the authors postulate that higher levels of LYS1 results in a more extensive hydrolysis of the PGN resulting in the loss of the correct sized fragment for induction of innate immunity. This seems a reasonable hypothesis but one that requires experimental validation. This requires the use of a quantitative assay for the immune response that can be followed over time. There are many such relatively easy assays that could be used, such as ROS production or gene expression. The authors should test whether the* LYS1 *OE and KD lines differ in their temporal response to both PGN and bacterial inoculation using such assays. If their hypothesis is correct, then one might predict that the OE lines would show some response very early but this would decline rapidly as the elicitor is completely hydrolyzed. One might also predict, for example, that adding higher levels of PGN elicitor might elicit a response on the OE lines (i.e., one might be able to saturate the enzyme and slow hydrolysis to the point where elicitation can be measured)*.

We agree that the results of the increased susceptibility of the *LYS1*^*OE*^ lines to *Pseudomonas* infection presented in Figure 9 (old Figure 8) are rather unexpected. However, there are other prominent examples in the literature where overexpression of a protein mimics the effect of genetic inactivation. For instance, strong overexpression of BAK1, a co-receptor of leucine-rich-repeat receptor kinases, triggers inappropriate plant cell death (Belkhadir et al., PNAS 2012, 109,297-302) as is also observed in *bak1* mutants (Kemmerling et al., Curr. Biol.17, 1116–1122). Hence, it can be assumed that wild-type levels of proteins are optimized during evolution and it is therefore not surprising that either too little or too much activity (at least in some cases) might have a deleterious effect.

However, as suggested by the Referees and as explained in more detail in Reply 3.7., we measured ion leakage in leaves infected with *Pto* DC3000, but could not observe any differences in wild type plants compared to *LYS1*^*KD*^ or *LYS1*^*OE*^ lines. In all lines, ion leakage was increased starting at 2 to 3 hours post bacterial infiltration. Moreover, we analysed *FRK1* transcript levels at early time points, also using PGN concentrations up to 500 µg/ml, but again we could not observe an enhanced *FRK1* gene expression in the *LYS1*^*OE*^ lines at any time point. Notably, in our hands PGN does not induce an oxidative burst and was thus not used as an assay here.

As only *LYS1*^*OE*^ lines with a massively increased LYS1 protein level (lines 1 and 2) show an increased susceptibility to bacterial infection we believe it to be quite challenging to find the right time point for detecting differences in immune responses in *LYS1*^*OE*^ lines compared to wild type or *LYS1*^*KD*^ lines as the enhanced LYS1 enzyme activity is present at the beginning of all experiments. We might have simply missed the short time frame in which such differences might occur, if they do occur at all.

For further information, please refer to Reply 3.7.

*6) Association of PGN-degrading activity with LYS1. The authors must demonstrate that the PGN-degrading activity detectable upon over-expression of* LYS1 *is indeed due to LYS1 itself, and not to another enzyme whose expression or activity is increased by* LYS1 *over-expression. For this, the authors should transiently express a catalytically-inactive variant of LYS1 (based on the knowledge on similar enzymes in other kingdoms) and compare the PGN-degrading activity of the corresponding extract when compared to an extract from leaves over-expressing WT LYS1*.

As suggested, we generated Ala-replacement mutants for the two amino acid residues Asn154 and Glu156, positions which were shown to be crucial for enzymatic activity of rubber tree hevamine (5). However, protein extracts from *N. benthamiana* leaves transiently expressing the LYS1^N^154^A/E156A^-myc or LYS1^N^154^A/E156A^-HA variants yielded the same enzyme activity towards 4-MUCT as the corresponding wild type extracts, hence these mutations did not render LYS1 catalytically inactive.

For hevamine a mutation of the D^125^E^127^ motif to A^125^A^127^ rendered the enzyme completely inactive. However, an amino acid alignment of hevamine and LYS1 indicated that LYS1 contains a N^154^E^156^ motif instead of the DE motif. Additional Aspartate residues are spread around this motif and any of those might be required for enzymatic activity. As we do not have any information from crystal structures for LYS1 it will be challenging to determine which Aspartate residue will be the most important one for catalytic activity.

Thus, our experimental data indeed cannot exclude the theoretical risk of an up-regulation of the activity of an enzyme other than LYS1 by LYS1 over-expression, but are there any precedents/examples in the literature showing such scenarios?

*7) What is LYS1 generating? In*
Figure 5*, the authors say that the mutants don't respond to the fragment generated, but the data shows there is some response. In fact, it looks like the basal FRK1 level with buffer is much lower in the mutants than in the Col-O. Therefore, are the fold inductions are retained and just the basal responses are altered? As mentioned above, all the data should be analysed together for significance groups. LYM1, LYM3 and CERK1 genes are essential for PGN signal transduction, so the mutants should have almost no response to soluble PGN. So it’s not clear whether this LYS1-generated peptidoglycan (PGN) fragments is soluble PGN or not, if* cerk1 *mutants are still responding to it*.

As requested, we now analyzed all data shown in new Figure 6 with ANOVA and the multiple comparison post hoc test. Following this analysis, only the wild type samples treated with supernatants from the digest with native LYS1 and mutanolysin are significantly different (p < 0.001) from all other samples. Additionally, the mutanolysin-digest was also significantly less active (p < 0.05) in the *lym1 lym3* double mutant. For a better understanding, Figure 6 now also includes the water control samples, which were set to 1 for each mutant.

However, we would like to point out that none of the PGN receptor mutants *lym1*, *lym3* and *cerk1* are completely devoid of any responses to PGN treatment, which we detected in our comprehensive micro-array studies (Willmann et al., PNAS 2011). This is most likely due to contaminations which are present in PGN preparations, and might not be removed by LYS1 digest and subsequent HPLC purification. Thus a null-response cannot be expected but, as demonstrated in Figure 9 (formerly Figure 8), a massive reduction in PGN responses in the mutants is evident.

Other issues to address:

Figure 1
*text says suspensions of Micrococcus luteus cell but legend and methods say it’s really cell wall prep/extract. It’s not clear what the negative control is. This should be documented in the Methods*.

We apologize for any confusion and now clearly state in the main text that *Micrococcus luteus* cell wall preparations were used.

To clarify what the negative control exactly is, the legend to Figure 1 now reads as follows:

“As negative control (nc) non-induced His6-tagged LYM3 bacterial lysates were used for affinity purification and eluates were subjected to turbidity assays.”

Also, we added this information in the Methods section:

“As negative control, a protein purification using non-induced cultures harbouring the His6-LYM3 construct was performed.”

Figure 1
*100 µg PGN but 100 µg/ml LPS. Need consistent description (per ml )*

Changed as requested.

Figure 2*: how can the authors be sure that the fast-migrating band corresponds to the full-length LYS1 protein, or actually, event to LYS1. Did the authors attempt to confirm the identity of this band by mass spectrometry?*

The fast-migrating band visible on the western blot in Figure 2 (now Figure 2 with the western blot replaced as requested in Inquiry 12) is only visible in the *LYS1*^*OE*^ lines, thus representing most likely LYS1. We agree with the referees that from the western blot analysis it cannot be judged if this fast-migrating band indeed represents the full-length LYS1 protein. Hence we purified LYS1 from the *LYS1*^*OE*^ line and analysed the purified LYS1 protein with MS analysis. For further information please also refer to Reply 12.

*In*
Figure 3*, the relative hydrolytic activity of LYS1 was calculated using chitinase as a standard. It would be better if the authors could directly measure LYS1 kinetic parameters (e.g., Kd). This would be an important measurement since it would get directly to the issue of physiological relevance. These are standard techniques, which should be within the authors' ability given their access to purified LYS1 protein*.

We agree with the Referees and now indicate specific enzyme activities as calculated in comparison to the commercially available chitinase and lysozyme, for which activities are now also shown (new Figure 4).

We also determined K_m_ values for purified LYS1.

*In*
Figure 2*, there is another band which located between 35Kd and 40 Kd shown in LYS1OE samples, please clarify what it is. If it is possible, please do this experiment again; the picture looks very bad. What are the possible effects of proteolysis on the protein, which seems essential for its purification? To be thorough, the authors should identify the cleavage site so that they have a better idea of how much of the LYS1 protein remains. For example, could proteolysis be eliminating a key regulatory portion of the protein?*

We apologize for this low quality western blot and repeated this experiment (now Figure 2). Additional bands in the *LYS1*^*OE*^ line were most of the time hardly visible but most likely represent degradation products derived from the LYS1-GFP fusion protein.

As the analysis of the total mass of the cleavage product by MALDI was not possibly due to the large size of the protein with approximately 35 kDa, we analysed the major cleavage product by LC-MS/MS after tryptic in-gel digestion and by peptide mass fingerprint, analyses which were conducted in two independent laboratories (Boris Macek, Proteome Center Tübingen, and Hubert Kalbacher, Medical and Natural Sciences Research Centre, both University of Tübingen). As stated in Reply 10, this not only confirmed the identity of LYS1 in this band but also yielded peptides spanning almost the whole protein sequence. We lacked the first 53 amino acids at the N-terminus (with the signal peptide comprising 22 amino acids); however, as these analysis rarely yield a 100 % peptide coverage and as the far N- terminus might not ionize easily, we are confident that the cleavage of the LYS1-GFP fusion protein occurs between LYS1 and GFP yielding an untagged, full length LYS1 protein.

[Editors' note: further clarifications were requested prior to acceptance, as described below.]

*The authors did a good job trying to respond to the comments to improve the manuscript. They added analysis of an additional O/E line that does not seem to cause statistically significant increase in* P. syringae *growth after 2 days; this line shows only moderate overexpression of* LYS1*. However, there are a few issues with part of the analysis that need to be addressed before final acceptance*.

*1) The authors need to harmonize the description of the statistical analysis with what is presented in the figures. The authors say they added multiple comparison tests, but the way they describe these is confusing. For example in*
Figure 4*, if the multiple comparison test was done, then all values in the panel should be labeled by grouping into like categories (i.e. A, B etc). Similarly in 6B (or 6C), significance at P<0.001 are given, but we cant tell whether LYS1 is different from mutanolysin or whether all the treatments in the mutants are giving the same or different values. If all the data was analyzed together, you should be able to label each group (all values that are similar get a different letter)*.

As requested, the data in Figures 4 and 6 are now presented as significance groups labeled with letters rather than asterisks.

*2) For the P. syringae infection experiments, the authors say in the discussion that overexpression or down-regulation of LYS1 causes the same magnitude of defect. Actually, they don't do any multiple comparison test of the data in*
Figure 9
*and it looks like the O/E are even more susceptible than the down-regulated plants. The authors should use multiple comparison test here to show whether the effects are the same or not. A couple of other things about the description of this experiment. (1) the title of the*
Figure 9
*should be changed, since there is no “mutation of LYS1”. (2) This experiment does not measure the growth rate since only one endpoint is taken. Therefore no conclusion about the rate can be given*.

We agree that the statement “We further found that plants overexpressing LYS1 were as susceptible as knock down mutants to bacterial infections, suggesting that defined LYS1 levels in wild-type plants are required for LYS1 immune function” in the Discussion was incorrect and misleading. We have therefore changed the statement into:

“We further found that plants overexpressing LYS1 were also susceptible to bacterial infections, suggesting that defined LYS1 levels in wild-type plants are required for LYS1 immune function.”

It was never intended to claim that *LYS1*^*OE*^ lines were AS susceptible as the *LYS1*^*KD*^ lines. The Referee is very much entitled to say that to state this would require more in-depth statistical analysis. However, we would like to state that by Student’s t-test, *LYS1*^*OE*^ lines as well as *LYS1*^*KD*^ lines were both more susceptible to bacterial infection than WT plants in a statistically significant manner. Student’s t-test is the most commonly used statistical analysis in current literature. We therefore prefer to represent our data using this statistical method.

As suggested, we changed the title of Figure 9 which now reads as follows: “Manipulation of LYS1 levels causes hyper-susceptibility towards bacterial infection and loss of PGN-triggered immune responses.”

We agree that for bacterial infections, no growth rates can be given. However, the figure legends state “bacterial growth” (and not growth rates), which can be determined by counting colony forming units at different times after inoculation.

*3) The authors tested the transcript level of receptors that would detect PGN fragments liberated by LYS1 in the* LYS1 *overexpression plants; this is important for excluding the possibility that high overexpression of* LYS1 *has unintended consequences on the receptor levels. A more informative experiment would be to check protein levels, as this is the most relevant assay (transcript levels don't always reflect protein levels). There is an available antibody for CERK1- did the authors try using this antibody?*

We agree with the Referees that protein levels are more informative than transcript levels. We have thus conducted a Western blot analysis with total protein from mature leaves using an anti-CERK1 antibody. New Figure 9—figure supplement 1 shows that there is no difference in CERK1 protein levels in WT and LYS1^OE^ lines.